# Pan-cancer ion transport signature reveals functional regulators of glioblastoma aggression

Alexander T Bahcheli[1,2,9], Hyun-Kee Min [2,3,4,9], Masroor Bayati [1,5], Hongyu Zhao [3,4,6], Alexander Fortuna[1], Weifan Dong [2,3,4], Irakli Dzneladze[1], Jade Chan[2,3,4], Xin Chen [3,4,7], Kissy Guevara-Hoyer [1,8], Peter B Dirks[2,3,4], Xi Huang [2,3,4 ✉] & Jüri Reimand [1,2,5 ✉]

## Abstract

**Ion channels, transporters, and other ion-flux controlling proteins, collectively comprising the "ion permeome", are common drug targets, however, their roles in cancer remain understudied. Our integrative pan-cancer transcriptome analysis shows that genes encoding the ion permeome are significantly more often highly expressed in specific subsets of cancer samples, compared to pan-transcriptome expectations. To enable target selection, we identified 410 survival-associated IP genes in 33 cancer types using a machine-learning approach. Notably, *GJB2* and *SCN9A* show prominent expression in neoplastic cells and are associated with poor prognosis in glioblastoma, the most common and aggressive brain cancer. *GJB2* or *SCN9A* knockdown in patient-derived glioblastoma cells induces transcriptome-wide changes involving neuron projection and proliferation pathways, impairs cell viability and tumor sphere formation in vitro, perturbs tunneling nanotube dynamics, and extends the survival of glioblastoma-bearing mice. Thus, aberrant activation of genes encoding ion transport proteins appears as a pan-cancer feature defining tumor heterogeneity, which can be exploited for mechanistic insights and therapy development.**

**Keywords** Cancer; Ion Channels; Glioblastoma; Neuron Projection; Target Discovery

**Subject Categories** Cancer; Membranes & Trafficking; Neuroscience

## Introduction

Developing rational cancer therapies is a challenge as broad-spectrum therapies fail to target tumor heterogeneity and multiple avenues of cancer progression (Dagogo-Jack and Shaw, 2018).

Molecular profiling as standard of care has advanced precision treatment regimens for many cancer types (Malone et al, 2020). Despite these advances, therapy development remains a long and uncertain process. Conversely, repurposing approved drugs is an appealing alternative with many successes (Zhang et al, 2020b), such as the use of the type II diabetes drug metformin as an anti-cancer agent (Saraei et al, 2019).

Ion channels permeate ions across membranes based on ionic electrochemical gradients. Voltage-gated ion channels are regulated by changes in transmembrane voltage potential and are involved in a variety of physiological processes, such as neuronal signal transmission and epithelial cell secretion. Ligand-gated ion channels are regulated by chemical messengers such as neurotransmitters at neural synapses and neuro-muscular junctions. Ion transporters actively move ions across membranes through energy consumption and conformational change. Gap junctions create intercellular connections to enable the passage of ions and small molecules between cells. Collectively, we refer to these proteins as the ion permeome (IP). The ion permeome is extensively studied in the context of human disease and includes many drug targets. For example, a common therapy for renal hypertension and cardiovascular disease involves $Ca^{2+}$ ion channel blockers (Hayashi et al, 2007; Triggle, 2006). IP inhibitors are frequently used as local anaesthetics, such as lidocaine and carbamazepine (Skerratt and West, 2015).

The ion permeome is emerging as a crucial regulator of tumorigenesis (Prevarskaya et al, 2018). Transient receptor potential (TRP) channels promote autophagy of kidney cancer (Hall et al, 2014), oxidative stress resistance in breast and lung cancer (Takahashi et al, 2018), and proliferation in esophageal and HRAS-driven cancers (Jung et al, 2019; Shi et al, 2009). Mitochondrial ion channels have been targeted to induce cancer apoptosis (Leanza et al, 2017; Yagoda et al, 2007), while the ORAI family of store-operated calcium channels are involved in prostate cancer oncogenesis (Dubois et al, 2014). Breast cancer metastasis is orchestrated by TRP (Middelbeek et al, 2012), ORAI (Chantome et al, 2013; Yang et al, 2009), and potassium channels (Mu et al, 2003; Payne et al, 2022; Sun et al, 2016). The chloride channel CFTR and voltage-gated potassium channel KCNQ1

[1]Computational Biology Program, Ontario Institute for Cancer Research, Toronto, ON, Canada. [2]Department of Molecular Genetics, University of Toronto, Toronto, ON, Canada. [3]Developmental and Stem Cell Biology Program, The Hospital for Sick Children, Toronto, ON, Canada. [4]Arthur and Sonia Labatt Brain Tumour Research Centre, The Hospital for Sick Children, Toronto, ON, Canada. [5]Department of Medical Biophysics, University of Toronto, Toronto, ON, Canada. [6]Department of Neurosurgery and Hunan International Scientific and Technological Cooperation Base of Brain Tumor Research, Xiangya Hospital, Central South University, Changsha, Hunan, China. [7]Songjiang Research Institute, Songjiang Hospital, Shanghai Jiao Tong University School of Medicine, Shanghai, China. [8]Cancer Immunomonitoring and Immuno-Mediated Pathologies Support Unit, Department of Clinical Immunology, Institute of Laboratory Medicine (IML) and Biomedical Research Foundation (IdiSCC), San Carlos Clinical Hospital, Madrid, Spain. [9]These authors contributed equally: Alexander T Bahcheli, Hyun-Kee Min. ✉E-mail: Xi.Huang@Sickkids.ca; Juri.Reimand@utoronto.ca

were described as tumor suppressors in gastrointestinal cancers (Rapetti-Mauss et al, 2017; Than et al, 2014, 2016; Yamada et al, 2018). Recently, the sodium leak channel NALCN was identified as a regulator of gastric, pancreatic, and prostate cancer dissemination (Folcher et al, 2023; Rahrmann et al, 2022). In brain cancer, we found that potassium channel EAG2 and chloride intracellular channel CLIC1 promote medulloblastoma growth by regulating cell volume-gated mitosis (Francisco et al, 2020; Huang et al, 2012). EAG2 promotes medulloblastoma metastasis by reducing local cell volume at the trailing edge of migrating tumor cells (Huang et al, 2015). EAG2 and Kvβ2 form a potassium channel complex to regulate tumor-neuron interaction and promote growth, invasion, and chemoresistance of glioblastoma (GBM) (Dong et al, 2023). The mechanosensitive ion channel PIEZO1 senses tissue stiffness to drive GBM aggression (Chen et al, 2018). PIEZO2 regulates the blood-tumor barrier to modulate the chemotherapy sensitivity of medulloblastoma (Chen et al, 2023). Other studies further characterized ion channels in regulating tumor cell-intrinsic properties and cell-microenvironment interactions, establishing some as therapeutic targets in brain cancer (Hausmann et al, 2023; Hitomi et al, 2015; Huang and Jan, 2014; Mulkearns-Hubert et al, 2019; Ransom and Sontheimer, 2001; Turner and Sontheimer, 2014). Despite the reported roles of some ion channels in cancer pathways, the transcriptomic landscape and clinical significance of IP genes in human cancer has not been systematically explored to date.

Glioblastoma is the most common and deadliest form of primary brain cancer. Despite multi-modal therapy combining surgery, radiotherapy, and chemotherapy using the DNA alkylating agent temozolomide, median patient survival is only 15 months (Ostrom et al, 2022). GBM is characterized by genetic, molecular, and phenotypic inter- and intra-tumoral heterogeneity. Molecular subtypes of mesenchymal, proneural, and classical GBM have specific genomic mutations, gene expression signatures, and clinical characteristics (Brennan et al, 2013; Verhaak et al, 2010; Wang et al, 2017). Individual GBMs harbor diverse tumor and stromal cell populations. This tremendous degree of tumor heterogeneity drives therapy resistance and tumor recurrence (Lan et al, 2017; Meyer et al, 2015). As such, there is an urgent need to identify actionable therapeutic targets and treatment opportunities.

Here we analyzed the transcriptomic landscape and clinical associations of IP genes across 10,000 human cancer samples. We discovered that IP genes were excessively upregulated in subsets of tumors significantly more than expected, revealing a novel aspect of tumor heterogeneity. Using machine learning, we established a catalogue of IP genes whose elevated expression associated with patient survival outcomes. In GBM, we focused on two IP genes, *GJB2* and *SCN9A*, and demonstrated their roles in promoting GBM aggression using patient-derived tumor cells and xenograft models. Our study highlights alterations in the IP as a cancer hallmark and provides a useful resource for functional studies of IP genes for therapeutic and biomarker development.

## Results

### Transcriptomic landscape of the ion permeome in cancer

To interrogate IP in cancer, we analyzed 9352 cancer transcriptomes of 33 cancer types from the Cancer Genome Atlas (TCGA)

PanCanAtlas project (Cancer Genome Atlas Research et al, 2013) (Table EV1). We studied 276 high-confidence druggable IP genes from the Guide to Pharmacology database (Harding et al, 2022) (Fig. EV1; Dataset EV1). We first investigated pan-cancer expression of IP genes using dimensionality reduction and found clustering of cancer samples based on tissue and organ systems (Fig. 1A). For example, GBM and low-grade glioma (LGG) clustered together, as expected, as did kidney cancer subtypes (renal cell carcinoma (KIRC), renal papillary cell carcinoma (KIRP), kidney chromophobe (KICH)). Organ-specific clusters were also apparent, such as digestive tract-related cancers with colon (COAD), rectum (READ), pancreatic (PAAD), and stomach adenocarcinomas (STAD). Some cancer types also showed distinct clusters, such as skin cutaneous melanomas (SKCM) and uveal melanomas (UVM). Tissue- and disease-type specific patterns of IP gene expression suggest their roles in multiple cancer types.

Transcriptomic analysis revealed dramatic patterns of IP gene overexpression in individual cancer samples. First, we considered IP genes that were expressed in most samples of a given cancer type. A typical IP gene showed 10-fold upregulation in a minor subset of samples (9%) compared to other samples of the same cancer type (Fig. 1B). This affected 49 (18%) IP genes per cancer sample on average, based on non-parametric Tukey's outlier analysis (Tukey, 1977) (Fig. 1C). Widespread overexpression of IP genes was identified in most cancer types and in the pan-cancer cohort. Next, we considered the subset of IP genes with switch-like activation, defined as genes with zero expression in most cancer samples and non-zero expression in less than half of the samples of a given cancer type. Switch-like activation affected an additional 18% of IP genes on average (Fig. 1D,E; Appendix Fig. S1a). A typical IP gene was expressed in hundreds of copies in high-expression group of cancer samples (median 290 FPKM-UQ), while some genes exceeded these levels by several orders of magnitude ($10^4$–$10^5$ FPKM-UQ) (Fig. 1F).

To evaluate the significance of IP overexpression in cancer, we repeated the outlier analysis by re-sampling protein-coding genes (PCG) as controls. In all cancer types, overexpression of IP genes was significantly more pronounced compared to all PCGs (IP: 10-fold increase, 9% of samples *vs.* PCG: 3.7-fold increase, 5% of samples; $P < 10^{-6}$, permutation test) (Fig. 1B; Appendix Fig. S1b). We repeated the outlier analysis using two other major drug target classes: kinases and G protein-coupled receptors (GPCRs). Aberrant overexpression of IP genes significantly exceeded the overexpression of kinase genes. Interestingly, GPCR genes were also highly upregulated in most cancer types, although the extent and frequency of upregulation among IP genes was often even higher (Appendix Fig. S1). Outlier analysis of GBM subtypes also confirmed that overexpression patterns of IP genes exceeded PCGs, kinases, and GPCRs in classical, mesenchymal, and proneural GBMs (Fig. EV2A). In summary, IP genes undergo dramatic upregulation in a fraction of cancer samples, suggesting that these contribute to tumor heterogeneity and disease mechanisms.

### Survival associations of ion permeome genes in multiple cancer types

To study the importance of IP genes in cancer pathology, we systematically prioritized IP genes that significantly associated with patient survival in individual cancer types, using a machine

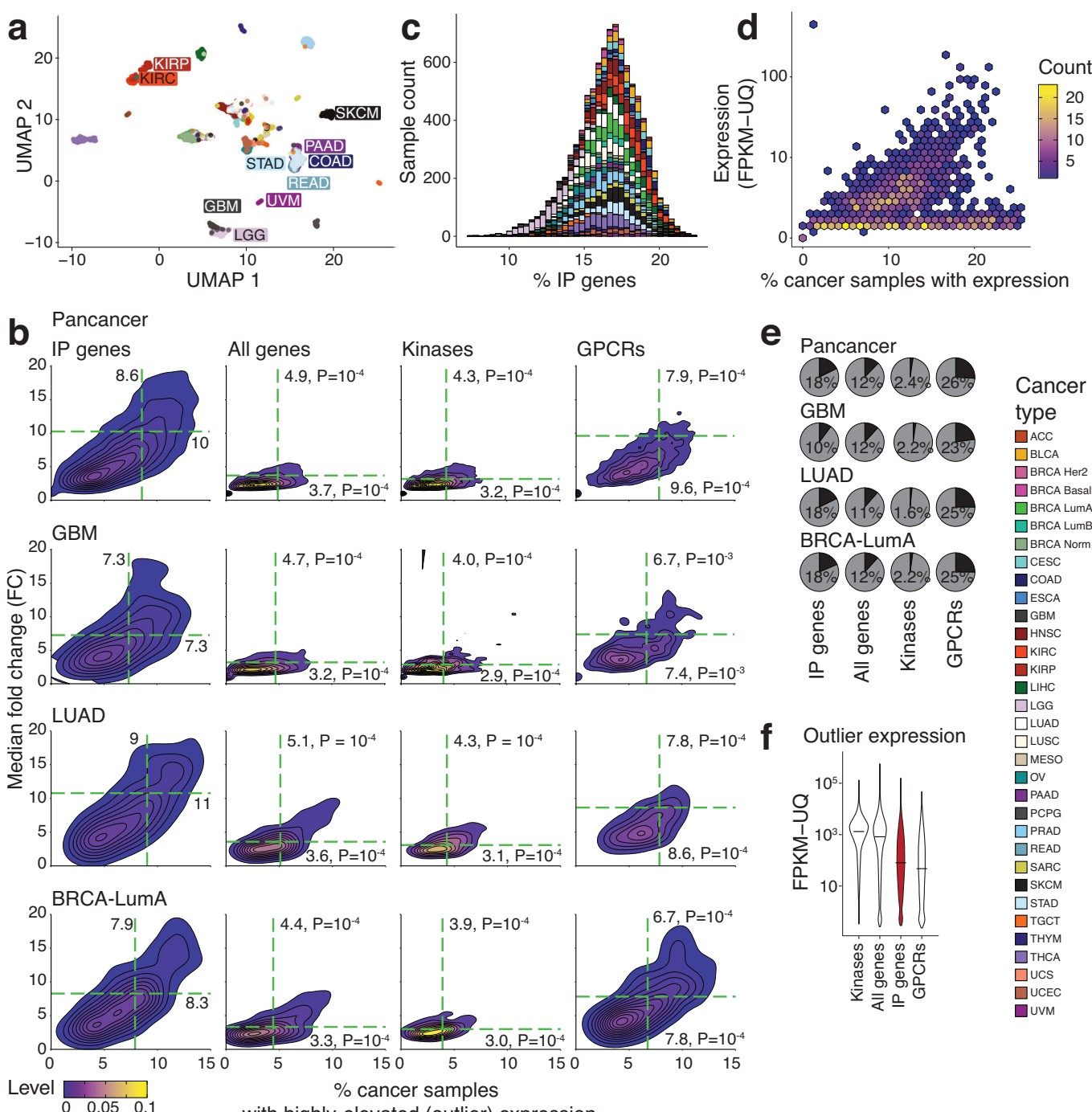

learning approach adapted from our previous study (Isaev et al, 2021). Briefly, we trained a set of Cox proportional-hazards (CoxPH) survival regression models on subsets of cancer samples with IP gene expression profiles as features, followed by regularisation that selected the most informative IP genes in each model (Fig. EV3). We nominated recurrently selected IP genes from the models as our top candidates. Cancer types were analyzed separately to identify IP genes, and either overall survival (OS) or progression-free survival (PFS) information was used as recommended previously (Liu et al, 2018) (Table EV1). We benchmarked

the analysis by randomly shuffling patient survival data, and as expected, found significantly fewer and weaker associations with IP genes, suggesting that our computational framework is appropriately calibrated (Appendix Fig. S2).

We identified 206 IP genes with 410 associations with patient survival in 33 cancer types (Fig. 2A,B). We found 12 IP genes per cancer type on average, while most IP genes with prognostic associations were found in only one or two cancer types (74%) (Table EV2; Dataset EV2), which is consistent with tissue-specific clustering of IP expression in cancer (Fig. 1A). Most survival

◀ **Figure 1. Highly elevated expression of ion permeome genes in cancer.**

(A) Dimensionality reduction analysis of cancer transcriptomes using ion permeome (IP) gene expression shows tissue-specific clustering in cancer. A UMAP projection of 276 IP genes in 33 cancer types from TCGA is shown. (B) IP genes are highly upregulated in subsets of cancer samples. Two-dimensional density plots show the joint distribution of gene expression increase (fold-change (FC), log2) and the fraction of cancer samples affected. IP genes (left) are compared to three gene sets as controls (middle to right): (i) all protein-coding genes, (ii) kinases, and (iii) GPCRs (i.e., two major classes of drug targets). Dashed green lines show median values. Control gene sets were down-sampled to IP gene counts and representative results with median FC are shown ($n = 10,000$). $P$-values from permutation tests are shown. (C) Histogram of IP genes with high expression in each cancer type. Fraction of IP genes with outlier expression for each cancer sample is shown. (D) Switch-like expression patterns of IP genes across all cancer types. 2D density plot shows the fraction of cancer samples with non-zero IP gene expression and the corresponding median non-zero expression values for each gene. (E) Pie charts show frequency of switch-like expression patterns among IP genes and all genes, kinases, and GPCRs as controls. (F) Expression levels of IP genes in the groups of cancer samples defined by outlier expression (red). Control gene sets (white) were down-sampled to IP gene counts over 10,000 iterations and representative results with median expression are shown. Source data are available online for this figure.

associations were found in prostate cancer and luminal-A breast cancer (23 and 24, respectively). Elevated IP gene expression associated with worse patient prognosis in most IP genes that we identified (240/410 or 59%). Several top-ranking IP genes were found in multiple cancer types, including *ACCN2, GRIN2D*, and *TRPV3* that associated with poor prognosis in six cancer types, and *P2RX6* in seven cancer types (Appendix Fig. S3a). *P2RX6* encodes a P2X receptor that increases renal cancer cell migration and invasion (Gong et al, 2019). *ACCN2* has been shown to promote tumor growth and metastasis in breast cancer (Gupta et al, 2016), and *GRIN2D* is an angiogenic tumor marker in colorectal cancer (Ferguson et al, 2016). *TRPV3* encodes a transient receptor potential cation-selective channel involved in temperature regulation pathways (Xu et al, 2002). Collectively, the catalogue of prognostic associations of the IP is a useful resource for functional studies and biomarker discovery.

## High *GJB2* or *SCN9A* expression associates with poor survival of GBM patients

We focused on IP genes in GBM, a fatal form of brain cancer with an unmet need for target identification. Four high-confidence IP genes were found, all of which were highly expressed in higher-risk patients: gap junction *GJB2* involved in non-syndromic hearing impairments (Snoeckx et al, 2005), voltage-gated sodium channel *SCN9A* involved in pain sensation in peripheral nervous system (Cox et al, 2006), aquaporin *AQP9* with roles in kidney cancer (Xu et al, 2019), and calcium-activated potassium channel *KCNN4* with roles in GBM (D'Alessandro et al, 2013; Grimaldi et al, 2016; Hausmann et al, 2023). Given its reported functions in GBM, *KCNN4* served as a positive control of our analysis. We confirmed the prognostic signals of these genes in multivariate analyses that accounted for patient age and sex (Fig. 3A). Besides GBM, *GJB2*, and *SCN9A* expression associated with poor prognosis in low-grade glioma, kidney, and uterine cancer (Appendix Fig. S3b), lending further confidence to these genes. We selected *GJB2* and *SCN9A* for further studies in GBM.

We examined *GJB2* and *SCN9A* expression in overall survival (OS) risk groups of GBM. For *GJB2*, GBM patients with high outlier expression had the worst prognosis, while for *SCN9A*, GBM patients with above-median gene expression associated with poor prognosis (Fig. 3B; Appendix Fig. S4a). For these genes, patient age was consistently highlighted as a prognostic covariate in our gene prioritisation. To study the age component in detail, we analyzed three age-based groups of GBMs and found strongest prognostic signals of *GJB2* and *SCN9A* in the middle age group (56–66 years, 51 patients), while the other age groups showed attenuated signals

(Appendix Fig. S4a). However, no significant survival differences were found in the middle age group compared to others, perhaps suggesting an interaction with the candidate genes (Fig. EV2D). The middle age group marks the greatest risk increase of presenting GBM, with twice the incidence rate compared to younger individuals and representing a third of the TCGA GBM cases within a decade of age (Ostrom et al, 2022).

We validated the survival associations of *GJB2* and *SCN9A* expression in two independent GBM cohorts, including 136 samples from the Glioma Longitudinal Analysis (GLASS) (Glass Consortium, 2018) and 55 samples from an earlier microarray dataset (Freije et al, 2004) (Fig. 3B). High *GJB2* or *SCN9A* expression associated with poor prognosis in both datasets based on median dichotomisation ($P < 0.1$; HR > 1.5), while weaker associations with highly elevated expression were also detected (Appendix Fig. S4b). When considering GBM subtypes or age groups separately, no significant survival differences for *GJB2* and *SCN9A* were found, except for an association with high *SCN9A* expression and worse prognosis in mesenchymal GBMs (Fig. EV2B). No significant associations between GBM subtypes and patient age groups were apparent (Fig. EV2C). Analyses of GBM subtypes have limited sample sizes and should be repeated in larger cohorts. Collectively, high expression of *GJB2* and *SCN9A* in high-risk GBMs implicate these genes as targets for functional experiments.

## *GJB2* and *SCN9A* expression is enriched in neoplastic cells and aggressive GBM subtypes

We characterized *GJB2* and *SCN9A* expression in GBM tumor regions and subtypes. First, we analyzed the anatomic transcriptional dataset of the Ivy GBM atlas (Puchalski et al, 2018) (Fig. 3C). Regions of microvascular proliferation showed reduced *GJB2* and *SCN9A* expression ($\log2FC < -2.9$, FDR $< 1.4 \times 10^{-3}$). These represent a GBM hallmark comprising both resident endothelial cells and differentiated malignant cells (Wang et al, 2010a). *GJB2* and *SCN9A* expression was lower in stromal fractions, which primarily include non-malignant fibroblasts (Clavreul et al, 2014). Higher *GJB2* expression appeared near the necrotic centers of GBMs in pseudo-palisading and peri-necrotic zones. Thus, *GJB2* and *SCN9A* are downregulated in anatomical regions with fewer tumor cells, while *GJB2* is upregulated in highly proliferative and motile regions of GBMs.

Next, we studied *GJB2* and *SCN9A* in transcriptomic and methylation subtypes of GBM (Brennan et al, 2013; Colaprico et al, 2016) (Fig. 3C). *GJB2* expression was higher in mesenchymal GBM and related methylation subtype class-1 ($\log2$ FC > 1.2, FDR < 0.05) (Brennan et al, 2013). Patients with mesenchymal GBM have worse

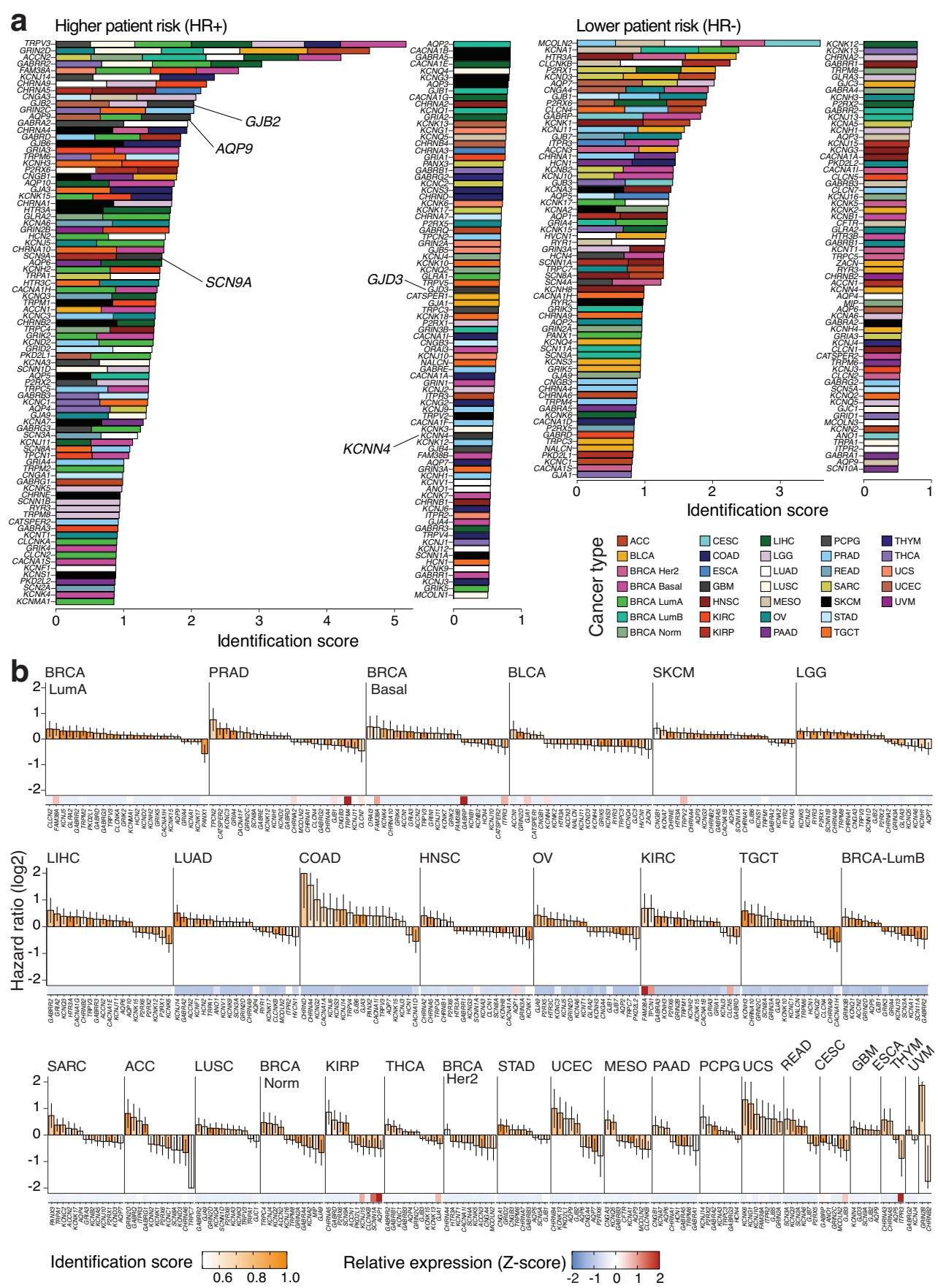

**Figure 2.  Patient survival associations of IP genes in cancer.**

(A) 206 IP genes with 410 patient survival associations in 33 cancer types, prioritized by their detection frequency in our elastic net framework (*X*-axis). IP genes associated with patient survival in glioblastoma (GBM) are labeled. (B) Catalogue of survival-associated IP genes in individual cancer types. Bar plots show median univariate hazard ratios (HRs) and error bars show 95% confidence intervals (top). Tile plots show median expression values of IP genes (bottom). Z-transformed relative expression values of individual IP genes compared to all protein-coding genes are shown. Sample sizes of HRs are shown in Table EV1. Source data are available online for this figure.

prognosis due to highly infiltrative and aggressive tumors (Verhaak et al, 2010). Lower *SCN9A* expression was found in classical GBM based on both classifications. Analysis of genomic alterations showed that *SCN9A* expression was associated with chromosome 19 and 20 co-gains, which are found in many classical and some mesenchymal GBMs (Brennan et al, 2013). Lower *SCN9A* expression associated with *MGMT* promoter methylation, an indicator of therapeutic response to temozolomide treatment (Rivera et al, 2010).

We then determined the cell types expressing *GJB2* and *SCN9A* using single-cell transcriptomics datasets of *IDH1/2* wildtype GBMs (Neftel et al, 2019) and *IDH1/2*-mutant and -wildtype GBMs (Johnson et al, 2021) (Fig. 3D). Both genes were expressed in subsets of neoplastic cells while their expression was undetectable in non-cancer cells, collectively showing an enrichment towards cancer cell fraction in both studies ($P < 10^{-6}$, Fisher's exact test). Among neoplastic cells, *GJB2* expression was higher in differentiated GBM cells and lower in proliferative stem-like cells. Higher *GJB2* expression in differentiated cells was characteristic of *IDH*-wildtype GBMs with worse prognosis (Johnson et al, 2021), while reduced *GJB2* expression and enrichment of stem-like cells was apparent in *IDH*-mutant gliomas with improved prognosis (Cohen et al, 2013). Compared to *GJB2*, *SCN9A* expression was more uniform across neoplastic cell types. Besides these, *GJB2* was expressed in myeloid cells while *SCN9A* was expressed in macrophages. Other non-cancer cells showed little or no expression of the two genes.

We compared *GJB2* and *SCN9A* expression in normal brain samples relative to their expression in GBMs using Tukey's outlier analysis (Fig. 3E). Expression profiles of 13 types of normal brain samples from 339 individuals were retrieved from the Genotype-Tissue Expression (GTEx) project (GTEx Consortium, 2013). As expected, the GBMs we classified as outliers showed significantly higher expression of *GJB2* and *SCN9A* than normal brain samples: *GJB2* ranked among the 24% most highly expressed genes in the outlier GBM group, while it ranked much lower in normal brain tissues and non-outlier GBMs (i.e., in top 70%; $P = 1.5 \times 10^{-13}$, Mann–Whitney *U*-test). Similarly, *SCN9A* expression was significantly higher in outlier GBMs compared to non-outlier GBMs and normal tissues (45% vs. 75%, $P = 1.1 \times 10^{-9}$). In contrast, *GJB2* and *SCN9A* expression in non-outlier GBMs was comparable to normal brain tissues.

Lastly, we studied genomic alterations and DNA methylation of *GJB2* and *SCN9A* in TCGA datasets (Fig. EV4). We observed no associations with patient survival and DNA promoter methylation, protein-coding mutations, or copy number alterations (CNAs) (Fig. EV4A). Only few protein-coding SNVs were found: *SCN9A* mutations were found in six GBMs while *GJB2* had none (Fig. EV4C). Interestingly, we found a co-occurrence of CNAs affecting *SCN9A* and *IDH1* ($P = 6.23 \times 10^{-23}$, Fisher's exact test) as the two co-located genes were often affected by the same CNAs in 20/145

GBM samples (Fig. EV4D). While *IDH1* is described through prognostic point mutations such as IDH1 R132H, the roles of *IDH1* CNAs or interactions with *SCN9A* are not established to date. Overall, the few genetic alterations of *GJB2* and *SCN9A* in GBM are likely not responsible for the high expression or patient survival associations of the genes and gene-regulatory or epigenetic mechanisms of activation are more plausible. Taken together, *GJB2* and *SCN9A* expression shows inter- and intra-tumoral heterogeneity in GBM. Their higher expression in malignant cell types and clinically relevant GBM subtypes implicate the functional significance of these two IP genes in GBM.

### *GBJ2* or *SCN9A* knockdown deregulates proliferative and neuron projection pathways

Next, we interrogated the functions of *GJB2* and *SCN9A* using shRNA-mediated knockdown in the patient-derived GBM cell line G729 (Meyer et al, 2015; Park et al, 2017). Transcriptome-wide profiling revealed dramatic changes induced by *GJB2* and *SCN9A* knockdown, with differential expression of 4647 and 2088 genes, respectively, including 640 commonly deregulated genes (absolute FC > 1.25, FDR < 0.05) (Fig. 4A; Appendix Fig. S5a). As expected, *GJB2* and *SCN9A* were significantly downregulated by knockdown (Fig. 4E). To interpret transcriptomic changes through patient GBMs in TCGA, we grouped patient GBMs by median *GJB2* or *SCN9A* expression and uncovered hundreds of differentially expressed genes (Appendix Fig. S5b).

To define the genes and pathways associated with *GJB2* or *SCN9A* knockdown, we jointly analyzed the four gene lists from patient-derived GBM cells and patient GBMs to identify integrative pathway enrichments using the ActivePathways method (Paczkowska et al, 2020). We discovered four major functional themes with differential expression: cell proliferation, neural and brain development, signal transduction pathways, and cytoskeletal and extra-cellular matrix processes, with 350 significant processes and pathways in total (*FWER* < 0.05; ActivePathways) (Fig. 4B; Table EV3). These pathways included cancer hallmarks of cell proliferation, cell cycle deregulation, DNA replication, and neural apoptosis, as well as signal transduction cascades such as the Wnt pathway. Cancer proliferation and invasion genes were downregulated, including proliferation marker gene *MKI67* with prognostic value in glioma (Torp, 2002), nerve growth factor receptor *NGFR* involved in GBM invasion (Ahn et al, 2016), and long non-coding RNA *MALAT1* with tumor suppressive function in GBM (Han et al, 2016) (Fig. 4E). The enriched pathways were supported by multiple transcriptional signatures, indicating that the target pathways of these two IP genes converge across our patient-derived GBM cells and patient GBMs.

We focused on a group of neuron projection processes that associated with both genes in our integrative pathway analysis. These included broader processes, such as regulation of neuron projection

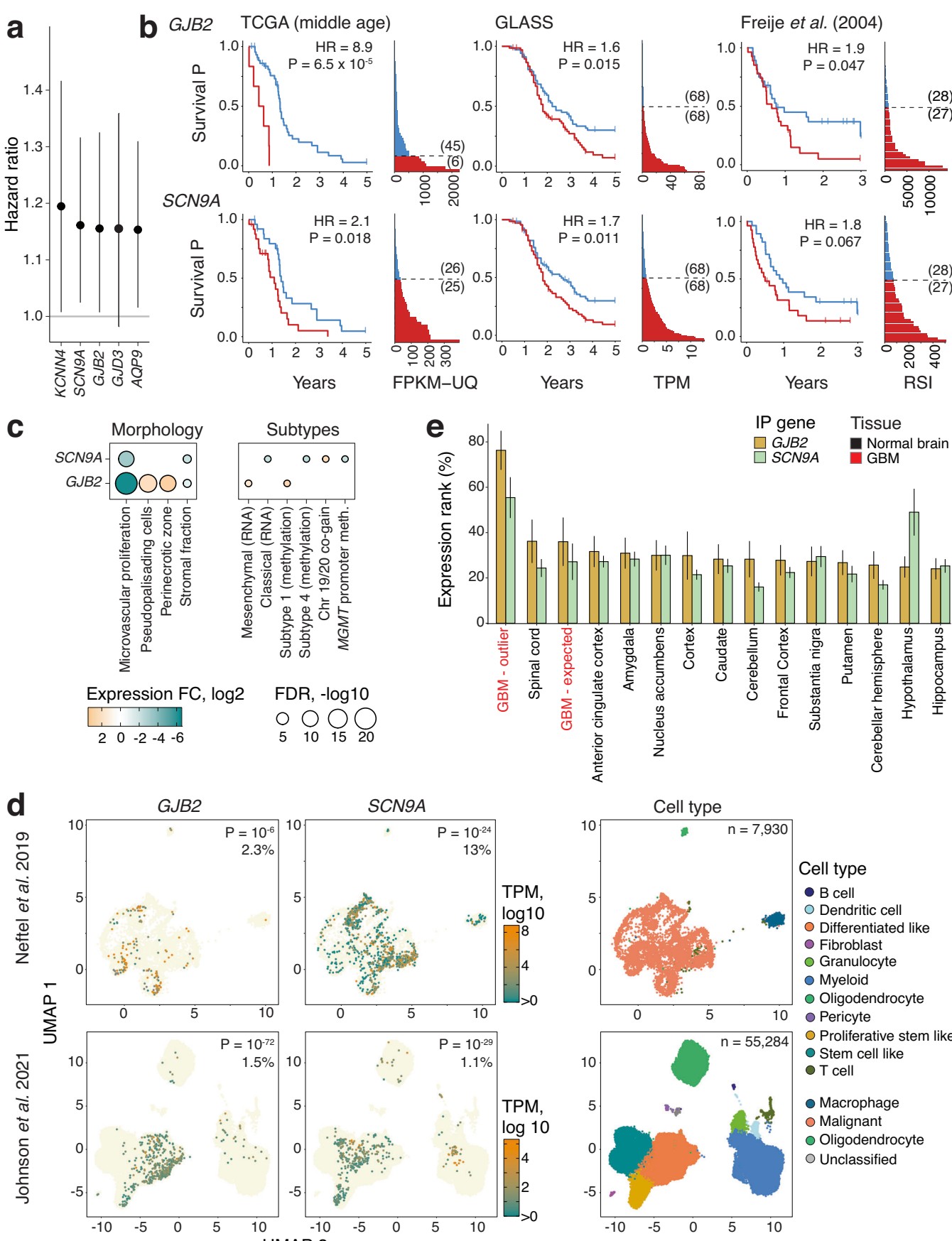

**Figure 3.   Survival associations and molecular features of *GJB2* and *SCN9A* in GBM.**

(A) Multivariate hazard ratios of IP genes were prioritized in 150 GBM samples. Error bars reflect 95% confidence intervals. (B) Kaplan-Meier plots of overall survival (OS) in GBM patients grouped by *GJB2* or *SCN9A* expression in TCGA (left) and two validation datasets (middle, right). Bar plots show gene expression in risk groups determined by Tukey outlier analysis for *GJB2* and median dichotomisation for *SCN9A*. Wald *P*-values, Univariate HR values, and sample counts are shown. Patient age in TCGA was included as a covariate. (C) *GJB2* and *SCN9A* expression associates with GBM subtypes and anatomical regions. FDR-corrected *P* values from Mann–Whitney *U*-tests are shown. (D) *GJB2* and *SCN9A* expression in single glioma cells from two single-cell RNA-seq datasets (Johnson et al, 2021; Neftel et al, 2019) shown as UMAP plots. Cells are colored by expression of *GJB2* (left) and *SCN9A* (middle). Cell type classifications were derived from the original studies (right). *P*-values from Mann–Whitney *U*-tests reflect enriched expression in malignant cell types and the fraction of total cells expressing *GJB2* or *SCN9A*. (E) Comparison of *GJB2* and *SCN9A* expression in GBMs and normal brain samples from GTEx. Quantile-normalized median expression values with confidence intervals are shown (±1 s.d). For GBM outlier groups, 13 samples for *SCN9A* and 18 samples for *GJB2* are included. All other groups include at least 100 samples. Source data are available online for this figure.

development (FDR = $1.4 \times 10^{-10}$) and specific enrichments such as axonogenesis and dendrite development (Fig. 4C; Table EV4). To prioritise individual genes in these pathways, we performed a complementary network analysis that examined gene interactions. We reconstructed a protein–protein interaction (PPI) network that captured 102 of the differentially expressed neuron projection genes, using interactomes from the BioGRID database (Oughtred et al, 2021) (Fig. 4D; Table EV5). The network highlighted *AKT1* and *PIK3R1* of the oncogenic PI3K/AKT signaling pathway involved in GBM (Li et al, 2016), and the tumor suppressor patched homolog 1 (*PTCH1*) (Stone et al, 1996). *PTCH1* was upregulated in both *GJB2* and *SCN9A* knockdowns, while *AKT1* and *PIK3R1* were deregulated in *SCN9A* knockdown (Fig. 4E). Thus, proliferation pathways appear to be deregulated by knockdown of our candidate genes.

Interestingly, we also uncovered genes and pathways regulating tunneling nanotubes (TNTs). TNTs are filopodia-like extensions between cells that enable cell-to-cell communication and promote tumor invasion, proliferation, and therapy resistance in GBM (Gimple et al, 2022; Rustom et al, 2004; Valdebenito et al, 2021; Wang et al, 2011). Our pathway and network analyses highlighted two Rac family small GTPases (*RAC1, RAC3*) that were down-regulated in *GJB2* knockdown cells, as well as the signaling adaptor *CDC42SE2* involved in TNT formation (Bid et al, 2013) (Fig. 4E). Furthermore, the PI3K/AKT signaling pathway differentially expressed in *GJB2* knockdown GBM cells is implicated in TNT (Wang et al, 2011). Collectively, transcriptome-wide signatures of *GJB2* and *SCN9A* indicate their roles in proliferative and neuron projection pathways in GBM. In particular, the TNT pathways deregulated in *GJB2* knockdown cells represent an intriguing avenue for further characterisation.

## GJB2 promotes tunneling nanotube formation, regulates filopodia dynamics, and increases GBM invasion

To define the role of GJB2 in TNTs, we investigated the cellular phenotypes of *GJB2* knockdown using three patient-derived mesenchymal GBM cell lines: G411, G729, and G797. First, we determined the impact of *GJB2* knockdown on Rho GTPase pathway genes. *GJB2* knockdown significantly decreased the expression of RAC Rho GTPase genes *RAC1*, *RAC2*, and *RAC3*, confirming our findings from transcriptomic and pathway analyses (Fig. 5A). Similarly, the TNT-associated signaling adaptor gene *CDC42SE2* was downregulated in *GJB2* knockdown cells, while tumor suppressor *PTCH1* was significantly upregulated. Furthermore, *GJB2* knockdown reduced RAC1 protein expression (Fig. 5B). Next, we monitored TNT dynamics in our patient-derived GBM cell lines (Fig. 5C). While GBM cells formed robust TNT networks

that connected different cells, we found a striking reduction in TNT lengths in all GBM cell lines upon *GJB2* knockdown (FC > 1.13, $P < 0.05$). These results were reproduced in GBM cells negative for an apoptosis marker (cleaved caspase-3), demonstrating that the shortened TNTs in *GJB2* knockdown cells were not caused by morphological changes from apoptosis (Appendix Fig. S6a). Since TNTs can be formed from physical interaction of two filopodia in double filopodia bridges (Chang et al, 2022) and RAC1 is a critical regulator of filopodia formation (Hall, 1998; Mehidi et al, 2019), we investigated the role of GJB2 on the dynamics of cell filopodia. Time-lapse imaging of membrane GFP (mGFP)-expressing G411 cells revealed that *GJB2* knockdown reduced the maximum extension length and lifetime of filopodia, while the extension rate and total number of filopodia remained unchanged (Fig. 5D). As the morphological changes from apoptosis take place within 2 h in vitro (Evan et al, 1992; Saraste and Pulkki, 2000), time-lapse imaging was performed over 2.5 h to ensure that the altered filopodia dynamics were not due to apoptosis. Among cells that did not exhibit apoptotic rounding or blebbing over the imaging duration, *GJB2* knockdown reduced the maximum extension length and lifetime of filopodia (Appendix Fig. S6b). Taken together, these results demonstrate that GJB2 regulates filopodia dynamics and TNT formation in GBM cells.

To characterize the role of GJB2 in GBM in vivo, time-matched tumors were collected from mGFP G411 xenografts treated with shRNA targeting *GJB2* or non-targeting controls. Immunofluorescence of RAC1 and phosphorylated myosin light chain 2 (pMLC2), a downstream effector of Rho GTPases, revealed that *GJB2* knockdown significantly diminished RAC1 and pMLC2 levels in tumors (Fig. 5E). These results corroborate our computational and in vitro findings that GJB2 is a critical regulator of RAC signaling. Since TNTs and filopodia facilitate tumor invasion (Jacquemet et al, 2015; Pinto et al, 2020), we examined the boundaries in *GJB2* knockdown and control tumors to analyze invasiveness. The size of invading tumor colonies as well as sinuosity, a measure of tumor infiltration, were significantly reduced in tumors with *GJB2* knockdown (Fig. 5F). We then examined filopodia, marked by mGFP, at the invasive front. *GJB2* knockdown tumors exhibited shortened filopodia length compared to control tumors (Fig. 5G). These results demonstrate that GJB2 promotes tumor invasion and regulates filopodia of GBM cells in vivo.

## GJB2 and SCN9A promote GBM growth in vitro and in vivo

Finally, we studied *GJB2* and *SCN9A* in the regulation of in vitro growth and in vivo tumorigenic potential of GBM cells. We studied the

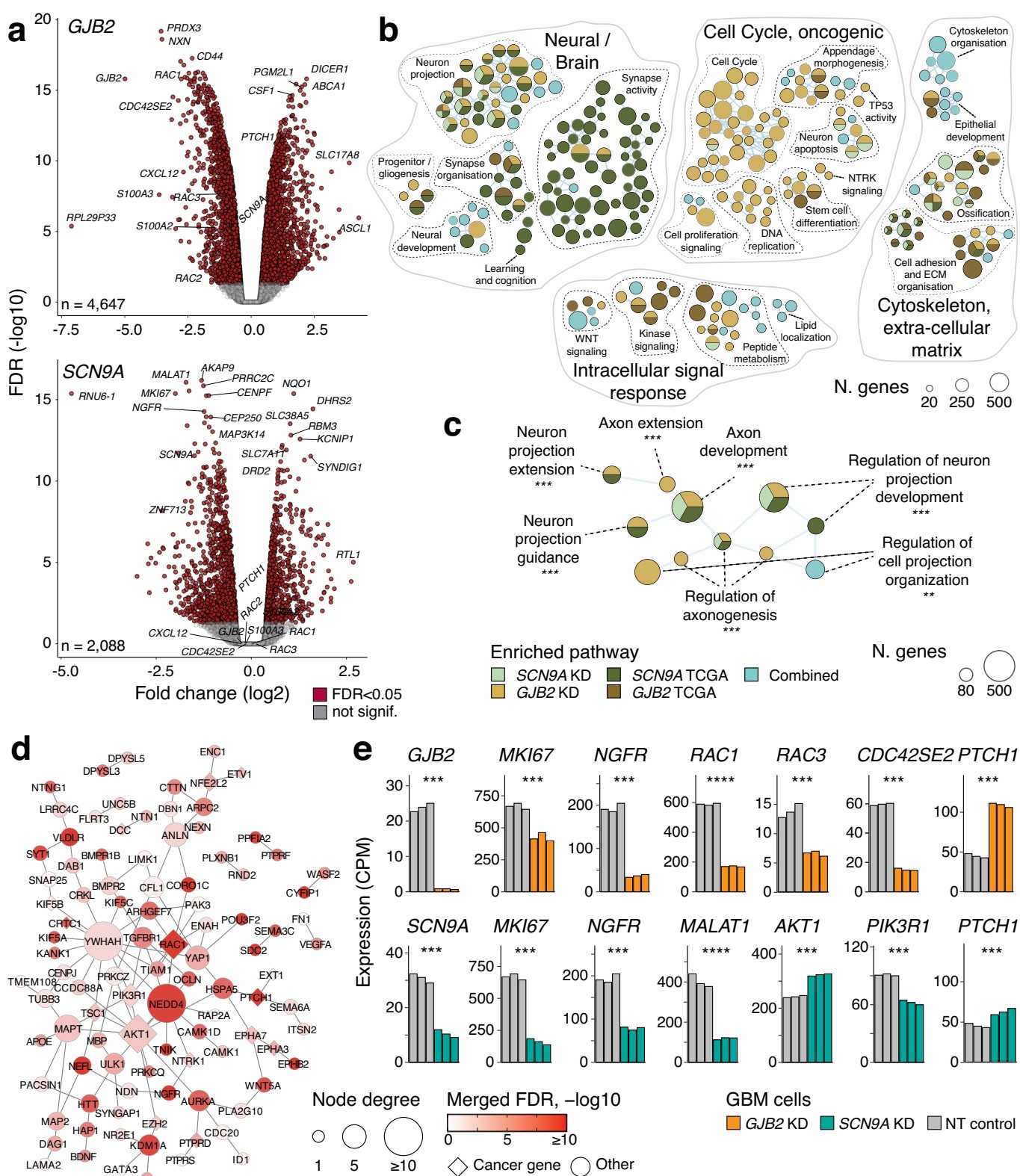

three patient-derived mesenchymal GBM cell lines G411, G729, G797 with high native expression of *GJB2* and *SCN9A* (Lan et al, 2017; Meyer et al, 2015; Park et al, 2017). Protein expression was validated by GJB2 and SCN9A immunostaining, which revealed that these were primarily expressed in the cell soma (Appendix Fig. S7a,b). First, we found that *GJB2* or *SCN9A* knockdown drastically reduced GBM cell viability (Fig. 6A). Second, we evaluated overall proliferative capacity using two approaches: by determining their sphere forming frequency

**Figure 4. Transcriptomic profiling of GBM cells with *GJB2* or *SCN9A* knockdown (KD) indicates their roles in proliferative and neuron projection pathways.**

(A) Differentially expressed genes in patient-derived GBM cells (G729, n = 3) with *GJB2* or *SCN9A* knockdown (EdgeR, FC > 1.25, FDR < 0.05). (B) Pathway enrichment analysis of genes associated with *GJB2* or *SCN9A* expression (ActivePathways, FWER < 0.05). Differentially expressed genes from knockdown experiments in GBM cells were jointly analyzed with genes differentially expressed in TCGA GBMs. The enrichment map shows enriched pathways as nodes that are connected by edges into subnetworks if the pathways share many genes. Network themes are labeled, and each pathway is colored by the transcriptomics dataset in which it was identified. (C) Subnetwork of neuron projection pathways from panel (B). Asterisks indicate statistical significance (*<0.05, **<0.01, ***<0.001, ****<10$^{-16}$). (D) Protein–protein interactions (PPIs) of genes in tunneling nanotube pathways. Nodes are differentially expressed genes in *GJB2* or *SCN9A* knockdowns in GBM cells from panel (A) and edges represent high-confidence PPIs from the BioGRID database. (E) Differential expression of selected genes involved in tunneling nanotube pathways, mitosis, and signal transduction. Normalized expression values (counts per million (CPM)) for *GJB2* or *SCN9A* knockdown and non-targeting (NT) controls are shown (n = 3). FDR values from EdgeR are shown as asterisks. Source data are available online for this figure.

using limiting dilution assay (LDA) (Brix et al, 2021; Hu and Smyth, 2009; Ploemacher et al, 1989), and by examining mitotic index using immunostaining of the mitotic marker phospho-histone H3 (pHis3). *GJB2* or *SCN9A* knockdown effectively abolished sphere formation (Fig. 6B,C) and significantly reduced the percentage of cells undergoing mitosis (Fig. 6D). Third, we investigated the roles of GJB2 and SCN9A in regulating GBM growth in vivo (Fig. 6E,F). We orthotopically injected luciferase-expressing G411 cells into immunodeficient NOD-SCID gamma mice. We monitored the survival of GBM-bearing mice and examined tumor growth using non-invasive bioluminescence imaging. *GJB2* or *SCN9A* knockdown markedly reduced tumor growth. Mice bearing GBMs with either *GJB2* or *SCN9A* knockdown displayed significantly prolonged survival ($P < 0.05$, Wald test). Examination of GJB2 and SCN9A by immunofluorescence in endpoint tumors revealed comparable expression in *GJB2* or *SCN9A* knockdown tumors and control tumors (Fig. EV5), suggesting that the mice in the knockdown groups succumbed to shRNA-escaper tumors. Collectively, these results demonstrate that GJB2 and SCN9A promote GBM growth in vitro and in vivo, are consistent with the findings that GBM-relevant genes and pathways are altered by their deficiency (Fig. 4) and establish *GJB2* and *SCN9A* as functional regulators of GBM aggression.

## Discussion

Ion channels and other IP genes comprise a large class of drug targets. More than 15% of FDA-approved drugs target ion channels to treat a variety of human diseases, such as diabetes, hypertension, and neurological disorders (Hayashi et al, 2007; Skerratt and West, 2015; Triggle, 2006). However, the expression patterns and functional roles of IP genes are understudied in cancer. In this study, we found that IP genes encoding ion channels, ion transporters, and gap junctions are highly upregulated in subsets of cancer samples at frequencies and magnitudes that significantly exceed most protein-coding genes. This is apparent in most major cancer types, thereby revealing a fundamental characteristic of cancer. On average, each cancer sample shows elevated expression of dozens of IP genes at levels significantly exceeding their physiological range. Our machine learning analyses provide a catalogue of IP genes that correlate with patient survival in cancer and have versatile roles in bioelectrical signaling. This is a useful resource for identifying biomarkers, validating therapeutic targets, and repurposing approved drugs that act on the ion permeome.

*GJB2* and *SCN9A* are implicated in monogenic diseases with emerging implications in cancer. SCN9A functions in signal transduction in neurons including nociceptor pain signaling (Estacion et al, 2009). *GJB2* encodes a gap-junction protein

(connexin) whose autosomal recessive allele causes deafness in Asian populations (Barashkov et al, 2011). High *GJB2* expression is associated with worse prognosis in GBM and LGG, suggesting common mechanisms in high- and low-grade gliomas. *GJB2* and *SCN9A* have been linked to invasion and proliferation in prostate (Chen et al, 2019), lung (Campbell et al, 2013), gastric (Xia et al, 2016), and breast cancer (Liu et al, 2019), as well as metastasis (Ezumi et al, 2008; Naoi et al, 2007) and worse prognosis in several cancer types (Inose et al, 2009; Liu et al, 2019; Meng et al, 2022; Zhu et al, 2017). In GBM, however, the phenotypic and prognostic aspects of *GJB2* and *SCN9A* have not been characterized to date. We selected *GJB2* and *SCN9A* as high-priority target genes in GBM due to their associations with patient survival identified in our machine learning analyses. *GJB2* or *SCN9A* knockdown led to profound transcriptional dysregulation that disrupted proliferative and neuron projection pathways in patient-derived GBM cell lines. Notably, *GJB2* knockdown affected tunneling nanotube pathways that control intercellular communications within GBM. The lengths of TNTs and filipodia were disrupted by *GJB2* depletion in GBM cells, possibly via reduced RAC1 expression. We demonstrated that reduced expression of either *GJB2* or *SCN9A* strongly impaired GBM cell viability in vitro and in vivo. Further, both genes show intratumoral variation in GBM and are predominantly expressed in malignant cell types. Collectively, these data establish our top-listed genes as functional regulators of GBM aggression.

GBM networks are comprised of multicellular connections between tumor cells and neurons, astrocytes, and other cells of the tumor microenvironment that are indispensable for proliferation, invasion, metabolic rewiring, and therapy resistance (Osswald et al, 2015; Pinto et al, 2020; Venkataramani et al, 2022; Venkatesh et al, 2019; Zhang et al, 1999). Different connections have been characterized: tumor microtubes (TMTs) are over 500 μm in length, last for days, and consist of gap junction connections, whereas TNTs are shorter than 100 μm, last for hours, and are mostly open-ended with few connections (Venkataramani et al, 2022). GJB2 knockdown GBM cells showed reduced expression of RAC small GTPases (RAC1-3) and the CDC42 effector *CDC42SE2* that are involved in TNTs (Hanna et al, 2017; Zhang et al, 2020a), while no expression changes were found for previously identified TMT-regulating genes (Jung et al, 2017; Osswald et al, 2015), suggesting that GJB2 function may be specific to TNTs. Gap junction proteins, such as connexin 43 (*GJA1*), mediate intercellular electrical signaling through TNTs (Wang and Gerdes, 2012; Wang et al, 2010b). Electrical coupling of GBM cells through gap junctions is required for tumor growth (Venkatesh et al, 2019). Determining the precise subcellular localisation of GJB2 and its role in electrical conductance is needed to establish GJB2 as a direct

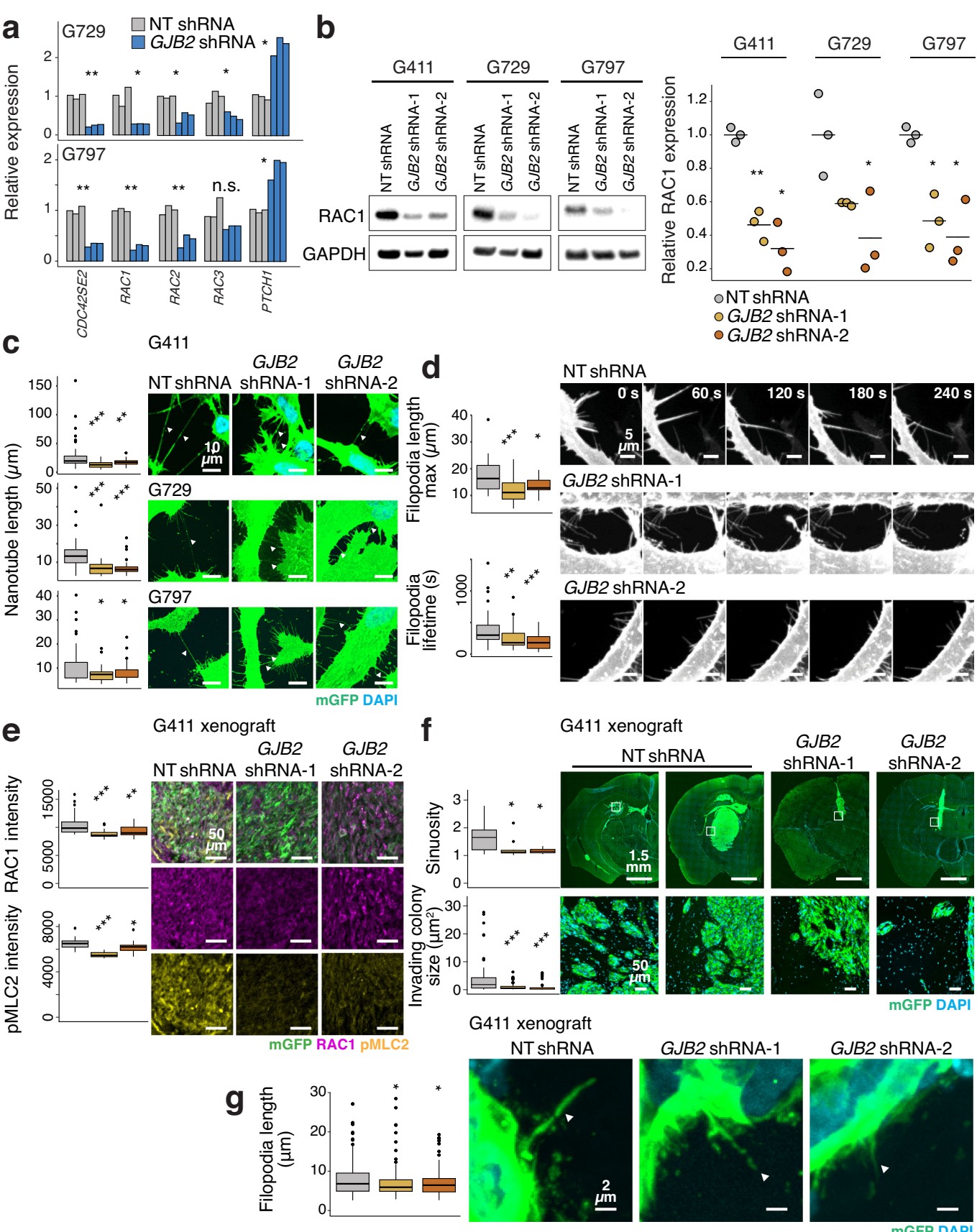

**Figure 5.** *GJB2* regulates TNT formation, filopodia dynamics, and invasion of GBM.

(A) TNT genes were dysregulated in *GJB2* knockdown cell lines. Comparison of mRNA expression of TNT pathway genes in *GJB2* knockdown GBM cells and non-targeting (NT) control cells measured using RT-qPCR ($n = 3$). Relative gene expression values were normalized to control genes. *P*-values were computed using Welch's *t*-tests and are shown as asterisks (*<0.05, **<0.01, ***<0.001, ****<10$^{-16}$). (B) Protein quantitation of RAC1 in *GJB2* knockdown cells. Western blot of RAC1 and the internal control GAPDH in three patient-derived cell lines targeted with two different *GJB2* shRNAs or NT controls. Relative expression of RAC1 in *GJB2* knockdown cells compared to NT controls from the western blots for three independent replicates (right). Horizontal lines display the mean relative protein expression in each group that was normalized in each cell line. *P*-values computed using Welch's *t*-tests are shown. (C) TNT projection lengths are reduced in *GJB2* knockdown cells (G411). TNT lengths (left) quantified from confocal microscopy images (right) of *GJB2* knockdown GBM cells labeled with mGFP and DAPI. Arrowheads indicate TNTs. (D) Timelapse images of mGFP tagged GBM cells. Filopodia extension length and filopodia lifetime in *GJB2* knockdown cells and NT control cells were measured for each cell at 30 s intervals over 180 intervals. (E) RAC1 and pMLC2 levels are reduced in *GJB2* knockdown tumors. Immunofluorescence imaging was performed in time-matched tumors collected 12 days post implantation. For xenograft experiments (E–G), results were derived from three mice for NT control and four mice for each *GJB2* shRNA. (F) GJB2 knockdown reduces tumor invasiveness. Boundary sinuosity and infiltrating tumor colony size were measured in confocal microscopy images of mGFP-expressing tumors. (G) *GJB2* knockdown tumors have reduced filopodia length. Data information: For cell experiments (A–D), all results represent at least three independent technical replicates. Two biological replicates (G729, G797) were used for (A) and three biological replicates (G411, G729, G797) for (B, C). For mouse xenograft experiments (E–G), results were derived from three biological replicates for controls and four biological replicates in each of the knockdown groups. Arrowheads in (C) indicate tunneling nanotubes and in (G) filopodia, which were visualized by tumor mGFP signal. *P*-values from Mann–Whitney *U*-tests are shown in (C–G). Box plots span the interquartile range (IQR; 25th–75th percentiles) where median values are shown as lines and whiskers reflect values within 1.5× of IQR. Source data are available online for this figure.

regulator of TNTs. We found that RAC Rho GTPases and actin regulators were downregulated in GBM cells upon *GJB2* knockdown. RAC1 is a major regulator of actin dynamics (Chang et al, 2022; Hall, 1998), and TNTs form from actin-rich membrane protrusions (Bid et al, 2013; Gimple et al, 2022; Pinto et al, 2020). Thus, our results suggest that GJB2 may also affect TNTs indirectly through RAC1 and other actin regulators. This is consistent with previous observations in HeLa cells, where GJB2 overexpression increased RAC1 activation (Polusani et al, 2016). Further studies are required to determine how GJB2 regulates RAC1.

In summary, we show that prominent activation of IP genes is associated with tumor heterogeneity and patient outcomes across the spectrum of human cancers. We present a catalogue of IP genes for future studies and establish two IP genes with oncogenic roles in GBM. The global IP gene alterations indicate that ionic flux-mediated bioelectrical signaling via aberrant ion permeome activity is a potential pan-cancer hallmark.

# Methods

## TCGA transcriptomics data and patient clinical information

Bulk tumor RNA-seq data and patient clinical information of the TCGA PanCanAtlas project (Cancer Genome Atlas Research et al, 2013; Data ref: The Cancer Genome Atlas, 2013) were collected from the Genomic Data Commons data portal. RNA-seq data in FPKM-UQ (fragments per kilobase million, upper quartile) were used unless specified otherwise. In cases with multiple tumor samples per patient, we selected the sample with the first barcode. Control samples and tumor samples lacking survival or RNA-seq data were removed. We analyzed cancer types with at least 50 samples, with 9352 samples of 29 cancer types in total, including 150 GBMs. Breast cancer (BRCA) subtypes were analyzed separately (luminal A, luminal B, *HER2* positive, basal-like, normal-like), using annotations from the R package *TCGABiolinks* (Colaprico et al, 2016; Data ref: TCGAbiolinks resource for TCGA data, 2016), resulting in distinct 33 cancer types. In gliomas (LGG), *IDH1/2* mutation status from *TCGABiolinks* was included as a covariate. All GBM samples from TCGA we analyzed were *IDH1/2* wildtype or unclassified. Additional analyses were conducted on subtypes of GBMs. Sample annotations of subtypes were

obtained from a previous study (Data ref: Wang et al, 2017; Wang et al, 2017) and the GBM samples were annotated to GBM subtypes based on TCGA sample barcodes. Nine GBM samples with missing subtype information were excluded. Data analysis was performed in Python (3.9.11) using custom scripts. Unless stated otherwise, statistical tests were performed using the *stats* package from the *Scipy* software library. The *ggplot2* R package was used for visualisations (R 4.1.3, *ggplot* 3.4.0).

## Ion permeome genes

Drug targetable ion permeome (IP) genes were retrieved from the Guide to Pharmacology (GtP) database (downloaded June 6, 2022) (Data ref: The IUPHAR/BPS guide to PHARMACOLOGY, 2022; Harding et al, 2022). IP genes included the classifications of voltage-gated ion channels (ICs), ligand-gated ICs, and other ICs. As controls, we studied two drug target families: kinases and G-protein coupled receptors (GPCRs). GPCRs were obtained from GtP. Kinases were retrieved from the UniProt database (pKin-Fam.txt, downloaded Sept. 15, 2022) (Data ref: UniProt: the universal protein knowledgebase, 2021; UniProt, 2021) and intersected with the list of enzymes in GtP. Genes lacking RNA-seq data in TCGA were excluded. In total, 276 IP genes, 391 GPCRs, and 505 kinases were included.

## Clustering cancer samples by IP gene expression

An unsupervised analysis of cancer samples using IP gene expression as features was performed using standardized, log1p-transformed FPKM-UQ expression values. The Uniform Manifold Approximation and Projection (UMAP) python package (McInnes et al, 2020) with default parameters was used for dimensionality reduction. Cancer samples were visualized in the first two UMAP dimensions and colored by cancer type.

## Highly elevated expression of IP genes

We identified IP genes with highly elevated expression using Tukey's outlier analysis (Tukey, 1977). Each cancer type and IP gene was analyzed separately. A cancer sample was considered to have highly elevated (outlier) expression of a given IP gene if its expression exceeded the 75th percentile of its expression across all

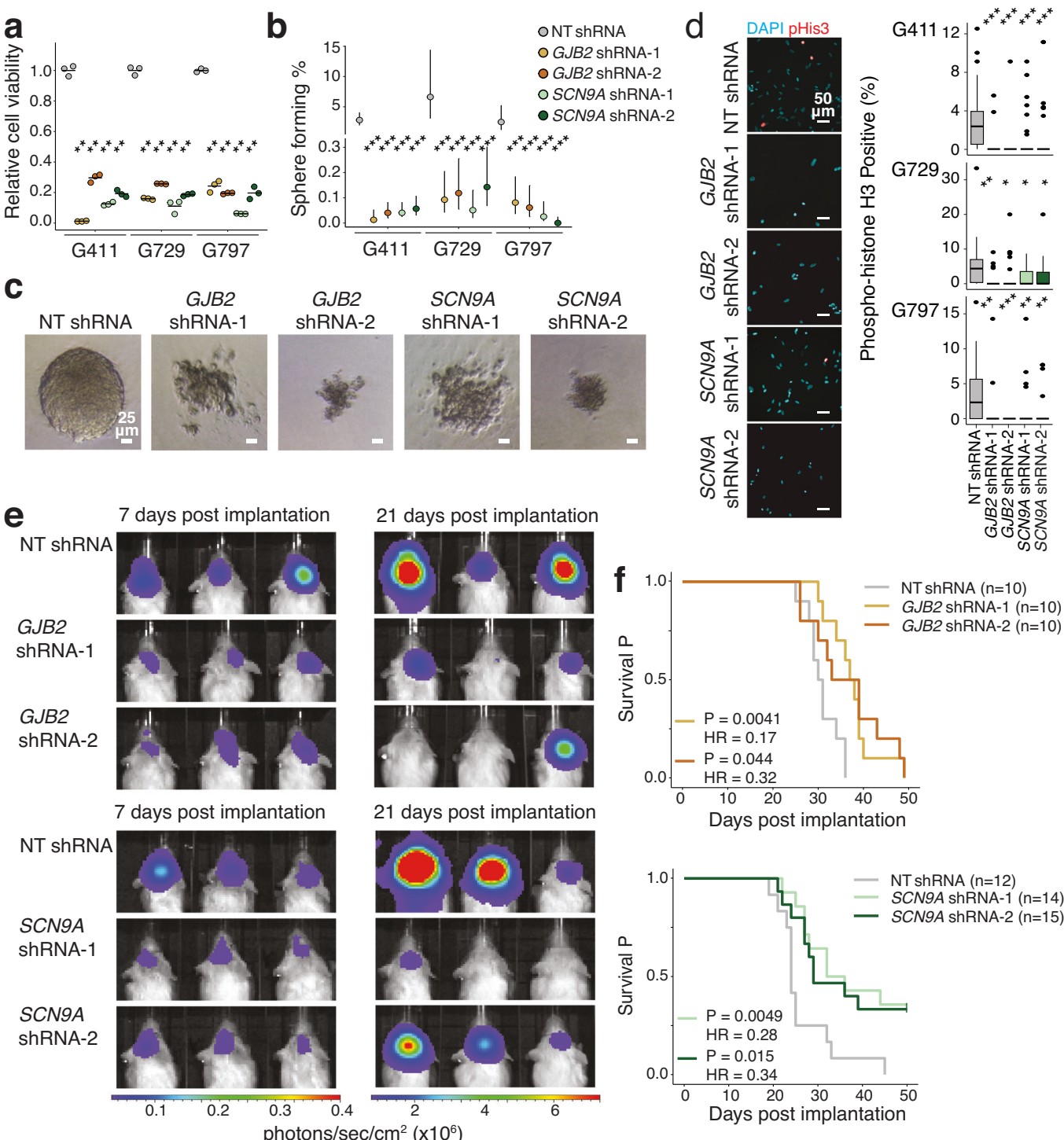

samples of the given cancer type by 1.5-fold the interquartile range (25–75%). Otherwise, the sample was classified as having an expected expression range. We computed expression fold-change (FC) values, comparing the cancer samples having highly elevated and expected expression of IP genes, as the ratio of median expression values of the two groups. Switch-like IP genes were annotated separately. Switch-like IP genes were defined as genes

that had zero expression in most samples (i.e., median zero) while fewer than half of the cancer samples had non-zero expression.

## Statistical analysis of elevated expression of IP genes

To evaluate the significance of elevated IP expression in cancer, we performed control analyses using (i) all protein-coding genes, and

◄  **Figure 6.   *GJB2* or *SCN9A* knockdown impairs GBM cell growth in vitro and in vivo.**

(**A**) *GJB2* or *SCN9A* knockdown reduces GBM cell viability in vitro. Cell viability was evaluated using an MTS assay in patient-derived GBM cell lines (G729, G797, G411). Horizontal bars show mean cell viability in each group, normalized to the mean of the controls (NT shRNA). For cell line experiments (A, B, D), all results represent at least three technical replicates and three biological replicates (G411, G729, G797). FDR-adjusted *P*-values from Welch independent *t*-tests are shown (*<0.05, **<0.01, ***<0.001, ****<10$^{-16}$). (**B, C**) *GJB2* or *SCN9A* knockdown reduces sphere formation of GBM cells. Brightfield imaging and limiting dilution assay (LDA) were performed on *GJB2* or *SCN9A* knockdown GBM cells and NT control cells. Sphere forming frequency was measured at the 14-day timepoint. Points show mean sphere formation frequency for each group of six technical replicates and the vertical lines show the full range of measurements calculated using the ELDA software. FDR-adjusted *P*-values from Welch-independent *t*-tests are shown (all FDR < 0.001). (**D**) *GJB2* or *SCN9A* knockdown reduces the mitotic activity of GBM cells in vitro. Mitotic activity was determined by immunostaining of mitosis marker phospho-histone H3 (Ser10). The percentage of phospho-histone H3-positive cells was quantified four days after lentiviral shRNA transduction. FDR-adjusted *P*-values from Mann–Whitney *U*-tests are shown. Box plots span the interquartile range (IQR; 25th–75th percentiles) where median values are shown as lines and whiskers reflect values within 1.5× of IQR. (**E**) *GJB2* or *SCN9A* knockdown impairs tumor growth in GBM xenografts in mice. Bioluminescence imaging was performed on mice following implantation of knockdown or NT control G411 GFP-luciferase cells. Radiance was measured 10 min after injection with 100 mg/kg luciferin on the IVIS Spectrum system. (**F**) *GJB2* or *SCN9A* knockdown improves mouse survival in GBM xenografts. Mice with gene knockdown xenografts or NT control G411 xenografts as controls were monitored for survival for 50 days. Survival analysis was performed independently for each shRNA and visualized with Kaplan–Meier plots. Biological replicates are indicated (*n*). Wald *P*-values and CoxPH HR values are shown. Source data are available online for this figure.

major classes of drug targets including (ii) GPCR genes, and (iii) kinase genes, as. The control gene sets were down-sampled with replacement to match the count of IP genes (276) over 10,000 iterations. Each cancer type was analyzed separately. Fractions of outlier cancer samples and median FC values from these iterations were used as reference to evaluate the cohort frequency and magnitude (fold-change) of IP gene upregulation in cancer. Cohort frequency and FC were visualized as 2D density plots for individual cancer types and the pan-cancer cohort. For control gene sets, representative iterations corresponding to median fold-change values were shown. *P*-values reflect the empirical probability of median outlier rates and median outlier FCs of the control gene sets exceeding the IP outlier rates and outlier FC.

## Identifying survival-associated IP genes

To find IP genes significantly associated with patient survival, we used a machine learning framework based on Cox proportional hazards (CoxPH) elastic net models and bootstrap analysis adapted from our previous work (Isaev et al, 2021). Log1p-transformed expression profiles of 276 IP genes were used as model features. Cancer types were analyzed separately, with a model response of either overall survival (OS) or progression-free survival (PFS), as recommended previously (Liu et al, 2018) (Table EV1). In each cancer type, IP genes with detectable expression were included (mean FPKM-UQ > 1 across all samples). IP genes were prioritized over 1000 iterations of elastic net survival regression models fit on random subsets of 80% of samples. At each iteration, feature pre-selection selected a subset of IP genes that associated with survival in univariate CoxPH regression (Wald test; *P* < 0.1). These were fitted using the Python package *CoxPHFitter* from the *lifelines* library. A multivariate CoxPH model was then fitted with pre-selected genes as features and patient survival as a response. Clinical variables were also included as features to evaluate the complementarity of IP genes. These included patient age and sex, tumor grade and stage; and *IDH1/2* mutations for LGG (Cohen et al, 2013). We selected the best-performing penalty (α) using a grid-search with 5-fold cross validations using the *GridSearchCV* package from *sklearn*. Following elastic net regularisation, model features (i.e., IP genes and clinical variables) with non-zero coefficients were recorded. After all iterations, we selected the final IP candidate genes and clinical variables that were identified as features in the regularized CoxPH models (>50% iterations). To derive hazard ratios (HR) and 95% confidence intervals for the

selected IPs, univariate CoxPH models using all samples were used. Elastic net training, regularisation, and parameter evaluation were conducted using the *CoxnetSurvivalAnalysis* package from the *sksurv* library, with a fixed L1-ratio hyperparameter ($\lambda = 0.5$). Finally, to confirm that our approach was calibrated, we repeated the IP prioritisation workflow using 1000 simulated datasets generated by randomly shuffling patient survival outcomes while maintaining true IP gene expression profiles. We compared the results of these simulated datasets to the true datasets. As expected, simulated data revealed significantly fewer and lower-confidence IP genes compared to true datasets (Appendix Fig. S2).

## Additional survival analyses of IP genes in GBM

In GBM, we focused on the five prioritized IP genes that were further vetted using extended multivariate CoxPH models with patient age, sex, and *IDH1/2* mutations as features. Based on HRs and Wald *P*-values, we selected four IP genes (*GJB2*, *SCN9A*, *KCNN4*, *AQP9*) and excluded *GJD3* due to sub-significant survival association and HR in multivariate models. To evaluate survival associations, GBM samples were split into two groups using median-dichotomisation and outlier-based (Tukey) dichotomisation of IP expression. Survival associations were evaluated using CoxPH regression separately for the discovery data (TCGA) and two external validation datasets (see below). Survival associations of *GJB2* and *SCN9A* were the strongest in the middle age group of TCGA GBMs (55–66 years), potentially explained by the age variable that was the strongest clinical feature identified in our analysis. Kaplan-Meier plots were generated to visualize the survival differences between groups.

## External validation of survival associations

We used additional GBM transcriptomics datasets to validate the survival associations of *GJB2* and *SCN9A*. First, we studied 136 primary GBMs profiled in the Glioma Longitudinal Analysis (GLASS) Consortium (Data ref: The Glass Consortium, 2018; Glass Consortium, 2018), after excluding recurrent GBMs, duplicate samples per patient, and samples used in TCGA. All GLASS samples were *IDH1/2* wildtype. GLASS RNA-seq data were available as transcripts per million (TPM) units. Second, we used the microarray dataset (Data ref: Freije et al, 2004; Freije et al, 2004) (GEO accession: GSE4412) with 55 grade-IV gliomas for which *IDH1/2* mutation status was unavailable. Relative fluorescent units

(RFI) of gene expression were exponentially transformed to approximate normal distributions. In case of multiple cancer samples per patient, the alphabetically first sample was selected. We also performed survival analyses with covariates as described above. No significant associations with patient age were found, potentially due to smaller sample sizes or cohort composition.

## Candidate gene expression correlations with clinical, immune, and micro-environment features

To explore the potential roles of *GJB2* and *SCN9A* in GBM, we asked how their expression associated with molecular and pathological features of GBMs, including longitudinal expression profiles from the *GLASS* project (Glass Consortium, 2018) and anatomical expression data of the Ivy Glioblastoma Atlas (Data ref: Ivy Glioblastoma Atlas Project, 2018; Puchalski et al, 2018). The GBM subtypes based on DNA methylation and gene expression patterns were acquired from TCGA (Brennan et al, 2013; Data ref: Brennan et al, 2013). Immune cell infiltration and immune-based cancer subtypes of TCGA samples were obtained from Thorsson et al (Data ref: Thorsson et al, 2018; Thorsson et al, 2018). Molecular features of TCGA tumors, including recurrent somatic mutations, copy number alterations, and clinical subtypes were obtained from *TCGABiolinks* (Colaprico et al, 2016; Data ref: TCGAbiolinks resource for TCGA data, 2016). To associate features with the candidate genes, gene expression in samples with and without features were compared using two-tailed Mann–Whitney *U*-tests. For tumor subtypes and other multi-class features, we compared samples of a given subtype with samples of all other subtypes combined. Immune cell infiltration (ICI) profiles from CIBERSORT were first median dichotomized into two equal subsets of GBMs (high *vs*. low ICI). IP gene expression was compared between the resulting groups. Multiple testing correction was applied within each analysis and significant findings were reported (FDR < 0.05).

## Association of somatic genomic alterations and DNA methylation with patient survival

To study whether *GJB2* and *SCN9A* in GBM were also associated with somatic genomic or epigenomic alterations at *GJB2* and *SCN9A*, we analyzed gene promoter DNA methylation, genomic copy number alterations (CNAs), and somatic nucleotide variants (SNVs) with GBM patient survival information. For promoter methylation analyses, patients were median-dichotomized into high and low groups based on promoter methylation of *GJB2* or *SCN9A*. For CNAs and SNVs, patients were grouped based on the presence or absence of a copy number gain, loss, or non-silent SNV. Survival associations were evaluated using CoxPH regression with high promoter methylation status or mutation status provided as a binary feature in a univariate model. Kaplan-Meier curves were generated to visualize the survival differences between groups. DNA methylation, copy number data, and SNVs from TCGA (Cancer Genome Atlas Research et al, 2013; Data ref: The Cancer Genome Atlas, 2013) was downloaded using the TCGAbiolinks R package (May 9th, 2023) (Colaprico et al, 2016; Data ref: TCGAbiolinks resource for TCGA data, 2016). We obtained methylation data for 54 GBM samples, copy number data for 146 GBM samples, and SNV data for 150 GBM samples, thereby

limiting our survival analysis to GBM samples with matching expression, genomic, and DNA methylation data. Methylation data was available from the Illumina 450k platform as beta values measuring CpG site methylation. We limited the analysis to CpGs in the relevant gene promoters using CpG annotations available in the Human EpicV2 dataset (Data ref: Human EpicV2 dataset, 2017), which associates CpG probes with genes and functional genomic regions. For each gene, we calculated the mean beta value across the CpG probes in its promoter. The mean value was used as a proxy of promoter methylation and *GJB2* and *SCN9A* gene promoter methylation means were calculated. Copy number analysis included relative CNAs for each gene and copy number intensities for genomic segments. Relative copy number data for each gene was used to define CNAs: relative copy numbers of zero were considered balanced; relative CNAs above zero were considered gains, and relative CNAs below zero were considered losses. We investigated the correlations between *GJB2* and *SCN9A* and known GBM cancer-associated genes from the COSMIC Cancer Gene Census database (Data ref: COSMIC: the Catalogue Of Somatic Mutations In Cancer, 2019; Tate et al, 2019). We expanded the CNA analysis to include all GBM samples with CNA data available from TCGA. *SCN9A* CNAs for individual GBM samples were visualized by associating gene-level CNAs with the chromosomal segments of the same sample. Protein-coding SNVs *GJB2* and *SCN9A* were retrieved from the TCGA mutant allele frequency (MAF) file for GBM samples.

## Expression of GJB2 and SCN9A in tumors and normal brain tissues

To compare *GJB2* and *SCN9A* expression in tumors and normal tissues, we analyzed GBMs from TCGA and normal brain tissues from the Genotype-Tissue Expression (GTEx) dataset (version phs000424.v9, downloaded Oct. 5, 2022) (Data ref: The Genotype-Tissue Expression (GTEx) project, 2013; GTEx Consortium, 2013). Expression data (TPM) for 3326 samples from 399 patients were obtained from the GTEx data portal. For improved comparison, RNA-seq data in TCGA and GTEx were then rank-normalized across all protein-coding genes. Expression ranks of *GJB2* and *SCN9A* were compared between 13 types of GTEx brain tissues and two subsets of GBMs (i.e., GBMs with highly elevated outlier IP gene expression, and GBMs with expected expression). The mean expression ranks were shown with $+/-$ one standard deviation (s.d.) for each tissue type. Ranks of tissue types were compared using two-tailed Mann–Whitney *U*-tests and multiple testing correction (FDR) was applied.

## Expression of GJB2 and SCN9A in GBM in single-cell RNA-seq datasets

We studied *GJB2* and *SCN9A* expression in single-cell RNA-seq data of GBM samples from two studies: Neftel et al (2019) (Data ref: Neftel et al, 2019; Neftel et al, 2019) (7930 cells, *IDH1/2* wildtype GBMs) and Johnson et al (2021) (Data ref: Johnson et al, 2021; Johnson et al, 2021) (55,284 cells, *IDH1/2* wildtype, and mutant GBMs). Previously processed transcriptomics data and cell and phenotype type annotations were retrieved from the original studies. The UMAP method was applied to log1p-transformed TPM values. Cells were colored by expression of *GJB2* or *SCN9A*

(log10 TPM). The reference cell map used the cell-type annotations from the original studies.

## Patient-derived GBM samples and cell culture

GBM cells for functional experiments were obtained following informed consent from patients. Experiments were in accordance with the Research Ethics Board (REB approval #0020010404) at The Hospital for Sick Children (Toronto, Canada). Access to pathological data was obtained from the institutional review boards. Patient-derived GBM cell lines (G797 male, G729 female, G411 male), which were established from mesenchymal GBMs, were cultured using previously established protocols (Pollard et al, 2009) including serum-free NS cell self-renewal media (NS media) consisting of Neurocult NS-A Basal media, supplemented with 2 mmol/L L-glutamine, hormone mix (in house equivalent to N2), B27 supplements, 75 mg/mL BSA, 2 mg/mL Heparin, 10 ng/mL basic FGF and 10 ng/mL human EGF. GBM cell lines were grown adherently on culture plates coated with poly-L-ornithine and laminin and maintained in 37 °C tissue culture incubator with 5% $CO_2$. All cell lines were regularly checked for mycoplasma infections by DAPI staining.

## Knockdown of GJB2 and SCN9A in GBM cells

Knockdown experiments in GBM cell lines (G411, G729, G797) were performed using lentivirus-mediated shRNAs. Knockdowns were repeated for three replicates per cell line and knockdown efficacy of GJB2 and SCN9A was validated using RT-qPCR. Human pLKO.1 lentiviral shRNA target against GJB2 or SCN9A and pLKO.1-TRC-control vector were obtained from Dharmacon. Viral transduction was performed in antibiotics-free culture medium for 24 h. The following shRNA mature antisense sequences were used: GJB2 #1: GTCTTCACAGTGTTCATGATT; #2: GAACGTGTGC-TACGATCACTA. SCN9A #1: GCCCTCATTGAACAACGCATT; #2: GCTGATTTGATTGAAACGTAT.

## RNA-seq profiling of GJB2 and SCN9A knockdowns

RNA-seq data was generated in the patient-derived GBM cell line G729 with shRNA targeting GJB2, SCN9A, or non-targeting (NT) shRNA controls, as described above. Total RNA was collected 4 days post lentiviral shRNA transduction using RNeasy Plus Mini Kit (Qiagen #74134). Lentiviral transduction and RNA extraction were performed in triplicates. RNA integrity number (RIN) was determined using Agilent Bioanalyzer. All samples had RIN > 9.8. Library preparation was performed using NEBNext Ultra II Directional polyA mRNA Library Prep Kit. Sequencing was performed on Illumina NovaSeq 6000 with 30 million paired end reads per sample at 100 bp read length.

## Processing and data analysis of RNA-seq data of knockdown cell lines

We aligned RNA-seq reads to the reference human genome HG19 (GRCh37.p13) from the GENCODE, for better consistency with the TCGA dataset. Reads were mapped to the transcriptome using Rsubread (Liao et al, 2019) with default settings. Differential gene expression analysis of GJB2 and SCN9A knockdown cells was conducted on raw read counts. Three replicates treated with IC-targeting shRNAs for each IP gene were compared to three control replicates treated with NT shRNAs, using the EdgeR package in R (Robinson et al, 2010). First, lowly expressed genes were removed (mean count < 1). Second, we selected significantly differentially expressed genes with an absolute fold change of at least 1.25 (abs log2 fold change flc > 0.32) using the glmTreat method of EdgeR. P-values from EdgeR were corrected for multiple testing and significant genes were selected (FDR < 0.05). Individual gene expression values were visualized using counts per million (CPM).

## Transcriptomics analysis of TCGA data for GJB2 and SCN9A

To integrate the transcriptomics data from our knockdown experiments with patient tumor data, we detected the genes associating with GJB2 and SCN9A expression in the GBMs in TCGA. Raw RNA-seq counts were obtained from TCGABiolinks (Colaprico et al, 2016; Data ref: TCGAbiolinks resource for TCGA data, 2016) and lowly expressed genes were filtered (mean count < 1). TCGA GBM samples were split into two groups based on the median gene expression separately for GJB2 and SCN9A. Differential gene expression analysis was used to compare the groups using glmTreat from the EdgeR package (Robinson et al, 2010) and significant genes were selected (absolute FC ≥ 1.25, FDR < 0.05).

## Integrative pathway enrichment analysis

We performed an integrative pathway enrichment analysis to identify pathways and processes jointly associated with GJB2 and SCN9A expression in our cell line knockdown experiments and GBMs in TCGA. We used the ActivePathways method (Paczkowska et al, 2020) with a matrix of P-values representing differential expression estimates of all protein-coding genes in the four comparisons (i.e., GJB2 and SCN9A; both in cell lines and TCGA). Gene sets of biological processes from Gene Ontology and molecular pathways from Reactome were downloaded from the gProfiler web server (January 13, 2023) (Data ref: The g:Profiler web server, 2021; Reimand et al, 2007). Gene sets with 50–500 genes were used. All protein-coding genes measured in RNA-seq datasets were used as the background set. Genes with low expression (mean count < 1) were deprioritized prior to the analysis by setting their P-values to 1.0. The resulting pathways were corrected for multiple testing and significant results were selected using default settings (Holm family-wise error rate (FWER) < 0.05). The enrichment map of pathways and processes was created using the EnrichmentMap app in Cytoscape standard protocols (Shannon et al, 2003) (Reimand et al, 2019). The major functional themes were organized manually.

## Protein-protein interactions of neuron projection pathways

We constructed a PPI network of the differentially expressed genes in GJB2 and SCN9A knockdown experiments that were annotated in neuron projection pathways. First, we selected a subset of GO processes related to neuron projection from our pathway enrichment analysis (Table EV4). Of those pathways, we selected the genes that were differentially expressed in at least one knockdown

experiment. All human PPIs were downloaded from the BioGRID database (Data ref: The BioGRID database, 2021; Oughtred et al, 2021) (version 4.4.217, July 22, 2022) and filtered to include only high-confidence PPIs found in at least two studies (PubMed IDs). Self-interaction PPIs were excluded. The PPI network was then limited to the neuron projection genes defined above and visualized using Cytoscape (Shannon et al, 2003). Proteins in the network were prioritized by node degree (node size) and differential expression in knockdown experiments (node color; merged FDR values using Brown's method). We highlighted known cancer genes of the COSMIC Cancer Census database (Data ref: COSMIC: the Catalogue Of Somatic Mutations In Cancer, 2019; Tate et al, 2019).

## Mouse xenograft experiments

We performed orthotopic mice xenografts of patient-derived GBM cell lines using female NOD SCID gamma /J#5557 immunodeficient mice aged eight weeks. Mice were housed under aseptic conditions with filtered air and sterilized food, water, bedding, and cages. Animal procedures followed the Animals for Research Act of Ontario and the Guidelines of the Canadian Council on Animal Care, as approved by the Centre for Phenogenomics (TCP) Animal Care Committee (protocol 23-0288H). GBM cells (G411) were transduced with lentiviral vector pBMN (CMV-copGFP-Luc2-Puro, Addgene plasmid #80389, a gift from Magnus Essand) expressing GFP and firefly luciferase under the control of cytomegalovirus (CMV) promoter, or mGFP (FUmGW, Addgene plasmid #22479, a gift from Connie Cepko) expressing membrane-bound palmitoylated EGFP driven by UbC promoter. GFP+ cells were sorted by fluorescence-activated cell sorting (FACS). G411 GFP-Luc2 or mGFP cells were transduced with NT, *GJB2,* or *SCN9A* lentiviral shRNAs for 24 hours. Two days post transduction, cells were injected into mice. Mice were assigned NT or shRNA group at random. Mice were anesthetized using gaseous isoflurane and immobilized in a stereotaxic head frame. The skull of the mouse was exposed, and a small opening was made using sterile dental drill (Precision Guide) at 1 mm lateral and 2 mm posterior to bregma. At this location, cells were injected with a Hamilton syringe 2.5 mm deep at a rate of 1 μL/min using a programmable syringe pump (Harvard Apparatus). 2000 cells were injected for endpoint, and 10,000 cells were injected for time-matched experiments. For survival and bioluminescence imaging, *GJB2* knockdown and *SCN9A* knockdown xenografts were performed in separate batches. For *GJB2* knockdown, 10 mice were used for each of NT, shRNA #1, and shRNA #2 groups. For *SCN9A* knockdown, 12 mice were used for NT, 14 mice for shRNA #1, and 15 mice for shRNA #2. Kaplan-Meier curves were generated to compare the survival of mice in the groups. For time-matched tumor analysis, 4 mice were used for each of NT, *GJB2* shRNA #1, and shRNA #2 groups. Mice were euthanized 12 days post implantation. Brains were harvested and tumor engraftment was verified by fluorescence stereomicroscopy. Only the samples with engrafted tumors were used for analysis, which were 3 for NT and 4 for each of the shRNA groups. All procedures were carried out under sterile conditions.

## In vivo bioluminescence imaging

In vivo bioluminescence imaging was performed using the Xenogen IVIS Lumina System coupled with LivingImage software for data acquisition. Mice were anesthetized using gaseous isoflurane and imaged 10 min after intraperitoneal injection of 100 mg/kg luciferin.

## Cell viability, limiting dilution assay, tunneling nanotube, and filopodia imaging

Cell viability was determined using CellTiter 96 AQueous One Solution Cell Proliferation Assay (Promega), utilizing MTS reagent, which produces a colored formazan dye when metabolized by NAD(P)H-dependent dehydrogenase enzymes (Cory et al, 1991). GBM cells were plated at 1000 cells per well in poly-L-ornithine and laminin-coated 96 well plates and transduced with a 50-fold dilution series of lentiviral shRNA for 24 h. Seven days post transduction, CellTiter 96 AQueous One Solution Cell Proliferation Assay was performed according to manufacturer protocol. Formazan dye absorbance was read using a microplate reader (Molecular Devices). Differences in cell viability were tested using a two-tailed independent-sample *t*-tests. Limiting dilution assay was performed by plating GBM cells in a serial dilution ranging from 2000 to 3 cells per well on round-bottom 96-well plates in six technical replicates. The numbers of wells with spheres were quantified fourteen days after plating and data were analyzed by Extreme Limiting Dilution Analysis (ELDA) software (Hu and Smyth, 2009), which calculates the frequency of sphere forming cells and the differences between groups using Chi-square tests. GBM cells were transduced with lentiviral mGFP (FUmGW, Addgene #22479). GFP+ cells were sorted by FACS. mGFP-expressing GBM cells were transduced with lentiviral shRNA (MOI > 1). For imaging of tunneling nanotubes at 4 days post transduction, cells were fixed with 4% paraformaldehyde at room temperature for 20 min, then stained with DAPI. Images were acquired on Quorum Spinning Disk confocal microscope with 63x/1.4NA objective. For live imaging at 4 days post transduction, cells were imaged every 30 s for 90 min or 150 min. Tunneling nanotube length was quantified using PerkinElmer Volocity software (version 6.3.1). Filopodia length was quantified using ImageJ software (version 1.54d). Image file names were blinded for the analysis. Significant differences in nanotube measurements were tested for using two-tailed Mann–Whitney *U*-tests.

## Immunofluorescence staining

For cells, lentiviral shRNA-transduced mGFP GBM cells were cultured on poly-L-ornithine and laminin-coated coverslips. 4 days post transduction, cells were fixed with 4% paraformaldehyde at room temperature for 20 minutes, then permeabilized with 0.1% Triton X-100 (Bio Basic #TB0198) in PBS (Bio Basic #PD8117). Cells were blocked with blocking solution (0.1% Triton X-100, 10% goat serum (MilliporeSigma #G9023) in PBS) for 1 h, then incubated with primary antibody at 4 °C overnight and secondary antibody for 1.5 h at room temperature in the dark. Stained coverslips were mounted onto slides (VWR #48311-703) using Aqua-Mount (Lerner #13800). For tissues, tumor-bearing mice were intracardially perfused with PBS and ice-cold 4% paraformaldehyde. Harvested brains were cryoprotected in 30% sucrose and flash frozen in O.C.T. compound (Scigen #SGN4585), then cryosectioned at 12 μm thickness. Tissue sections were permeabilized with 0.1% Tween-20 (Bio Basic #TB0560) in PBS, then

incubated in blocking solution for 2 h at room temperature. Sections were then incubated in primary antibody at 4 °C overnight and secondary antibody for 2 h at room temperature. Stained sections were mounted with coverglass and Aqua-Mount. Primary antibodies were diluted in blocking solution, and secondary antibodies were diluted in blocking solution with 1 µg/mL DAPI. Primary antibodies used were rabbit anti-phospho-histone H3 (Abcam #AB5176, 1:1000), chicken anti-GFP (Aves Labs #GFP-1020, 1:200), mouse anti-RAC1 (BD Biosciences #BD610650, 1:100), rabbit anti-phospho-myosin light chain 2 (Cell Signaling #3671, 1:100), rabbit anti-cleaved caspase-3 (Cell Signaling #9661, 1:500), rabbit anti-Nav1.7 (Alomone Labs #ASC-008, 1:100), mouse anti-Cx26 (ThermoFisher #13-8100, 1:100), and mouse anti-STEM121 (Takara #Y40410, 1:100). Secondary antibodies used were Alexa Fluor 488 donkey anti-chicken IgG (#703-545-155), Rhodamine Red-X donkey anti-mouse IgG (#715-295-151), Alexa Fluor 647 donkey anti-mouse IgG (#715-606-151), Alexa Fluor 594 donkey anti-rabbit IgG (# 711-585-152), and Alexa Fluor 647 donkey anti-rabbit IgG (#711-605-152) from Jackson ImmunoResearch. All secondary antibodies were diluted 1:400 for cells and 1:200 for tissues.

## Acquisition and quantification of immunofluorescence images

All immunofluorescence images were acquired on the Quorum Spinning Disk confocal microscope. All quantification was performed on the Volocity software (version 6.3.1). For phospho-histone H3 positive percentage, randomly picked regions were imaged on the 10x/0.4NA objective. pHis3-positive nuclei were counted using the point counting tool. For tumor RAC1 and pMLC2 intensity quantification, randomly picked regions were imaged on the 10x/0.4NA objective. Tumor-specific RAC1 and pMLC2 signals were measured by using the object tool to create regions of interest around the tumor GFP signal, then quantifying RAC1 and pMLC2 intensity within these regions. For GJB2 and SCN9A imaging in cells, images were acquired with the 63x/1.4NA objective. mGFP signal was used to create regions of interest and measure GJB2 and SCN9A intensity within cells or filopodia. For tissues, images were acquired with the 20x/0.8NA objective. Tumor regions were specified by STEM121 signal or DAPI density. GJB2 and SCN9A intensities were measured in tumor regions. For quantification of invasiveness, whole brain stitches were created by imaging with the 10x/0.4NA objective. Infiltrating regions and tumor boundaries were imaged on the 20x/0.8NA objective. The 63x/1.4NA objective was used to image in vivo filopodia. Sinuosity was measured as boundary length divided by distance. Infiltrating tumor colony size and filopodia length were measured manually. Sample IDs were blinded for quantification.

## RNA extraction, reverse transcription, and RT-qPCR

Total RNA was collected 4 days post lentiviral shRNA transduction using GENEzol TriRNA Pure Kit (Geneaid #GZX200). RNA concentration was measured using NanoDrop 1000 Spectrophotometer, and 1 µg of RNA was reverse transcribed to cDNA using SensiFAST cDNA Synthesis Kit (Bioline #65054). qPCR reactions were set up using PowerUp SYBR Green Master Mix (Applied Biosystems #A25742) and real-time detection and quantification of cDNAs was performed on the Viia7 Cycler (Applied Biosystems) with 40 cycles of amplification. Viia7 System Software (Applied Biosystems) was used to determine Ct values with automatically set thresholds. Gene expression was normalized to *GAPDH* and analyzed using the ΔΔCt method. The following RT-qPCR primers were used (h, human): hGAPDH, 5′-CTC CTG CAC CAC CAA CTG CT-3′ (forward), 5′-GGG CCA TCC ACA GTC TTC TG-3′ (reverse); hRAC1, 5′-CGGTGAATCTGGGCTTATGGGA-3′ (forward), 5′-GGAGGTTATATCCTTACCGTACG-3′ (reverse); hRAC2, 5′-CAGCCAATGTGATGGTGGACAG-3′ (forward), 5′-GGAGAAGCAGATGAGGAAGACG-3′ (reverse); hRAC3, 5′-ACAAGGACACCATTGAGCGGCT-3′ (forward), 5′-CCTCGTCA AACACTGTCTTCAGG-3′ (reverse); hCDC42SE2, 5′-GGATCAG GAGACCTGTTCAGTG-3′ (forward), 5′-CCTTCGTATCCACGA GCTGCAT-3′ (reverse); hPTCH1, 5′-GCTGCACTACTTCAGA GACTGG-3′ (forward), 5′-CACCAGGAGTTTGTAGGCAAGG-3′ (reverse).

## Protein extraction and western blots

We quantified the protein levels of RAC1 in *GJB2* knockdown GBM cells. Total protein was extracted from multiple cell cultures (G411, G729, G797) using the RAC1 Activation Assay Biochem Kit per manufacturer's instructions (Cytoskeleton Inc, #BK035-S) five days post transduction with lentiviral non-targeting or *GJB2* shRNA. All protein lysates were homogenised for 20 min at 4 °C, then centrifuged at 4 °C and 14,000 rpm for 10 min. 10 mg of protein samples were resolved on a 10% Bis-Tris gel (Invitrogen, #NW00102BOX) at 200 V in MES running buffer (Invitrogen, #B0002). The proteins were transferred onto a PVDF membrane (Millipore, #IPVH0001) and blocked with 5% BSA in 0.1% Tween-20 in TBS. Membranes were incubated overnight at 4 °C in primary antibodies diluted in the blocking solution. Immunoreactive bands were visualized using HRP-conjugated secondary antibodies (Cell Signaling Technology), followed by chemiluminescence with ECL-plus Western Blotting Detection System (Amersham, #RPN2232). Chemiluminescence was imaged and analyzed using Molecular Imager VersaDoc MP4000 system (Bio-Rad). The primary antibodies used were mouse anti-Rac1 (1:500, Cytoskeleton Inc, #ARC03) and rabbit anti-GAPDH (1:3000, Cell Signaling #2118S). Experiments were performed in three biological replicates. Significant differences in protein expression were analyzed using two-tailed independent-sample *t*-tests.

## Statistical analyses of patient-derived cell lines and mice

No statistical methods were used to predetermine sample sizes in validation experiments. Statistical analyses were completed after the experiments without interim data analysis. No data points were excluded. All data were collected and processed randomly. Each experiment was successfully reproduced at least three times and the experiments were performed on different days.

# Data availability

RNA-seq data generated in this study are available from the ArrayExpress database with the accession number E-MTAB-12859.

## Peer review information

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

## Acknowledgements

We would like to thank Drs. Kieran Campbell and Sunit Das for constructive comments on the project. This work was supported by the New Investigator Award of the Terry Fox Research Institute (TFRI) to JR, the Canadian Cancer Society (CCS) Innovation Grant to JR and XH, the Canadian Institutes of Health Research (CIHR) Project Grants to JR and to XH, Investigator Award to JR from the Ontario Institute for Cancer Research (OICR), and Sontag Foundation Distinguished Scientist Award to XH. XH is a Catalyst Scholar at The Hospital for Sick Children and Canada Research Chair in Cancer Biophysics. ATB was supported by the Ontario Graduate Scholarship (OGS). HM was supported by OGS and the RestraComp Scholarship from The Hospital for Sick Children. All imaging was performed at the Imaging Facility at The Hospital for Sick Children. Funding to OICR is provided by the Government of Ontario. Some graphics were enhanced using the BioRender software. The results shown here are in whole or part based upon data generated by the TCGA Research Network: https://www.cancer.gov/tcga.

## Author contributions

**Alexander T Bahcheli:** Resources; Data curation; Software; Formal analysis; Validation; Investigation; Visualization; Methodology; Writing—original draft; Writing—review and editing. **Hyun-Kee Min:** Resources; Data curation; Formal analysis; Validation; Investigation; Visualization; Methodology; Writing—original draft; Writing—review and editing. **Masroor Bayati:** Data curation; Formal analysis; Investigation; Methodology. **Weifan Dong:** Investigation; Methodology. **Alexander Fortuna:** Data curation; Formal analysis; Investigation; Methodology. **Hongyu Zhao:** Investigation; Methodology. **Irakli Dzneladze:** Data curation; Investigation; Methodology. **Jade Chan:** Data curation; Investigation; Methodology. **Xin Chen:** Data curation; Investigation; Methodology. **Kissy Guevara-Hoyer:** Data curation; Formal analysis; Validation; Investigation; Visualization. **Peter B Dirks:** Resources. **Xi Huang:** Conceptualization; Resources; Supervision; Funding acquisition; Investigation; Project administration; Writing—review and editing. **Jüri Reimand:** Conceptualization; Resources; Supervision; Funding acquisition; Investigation; Writing—original draft; Project administration; Writing—review and editing.

## Disclosure and competing interests statement

The authors declare no competing interests.

# Expanded View Figures

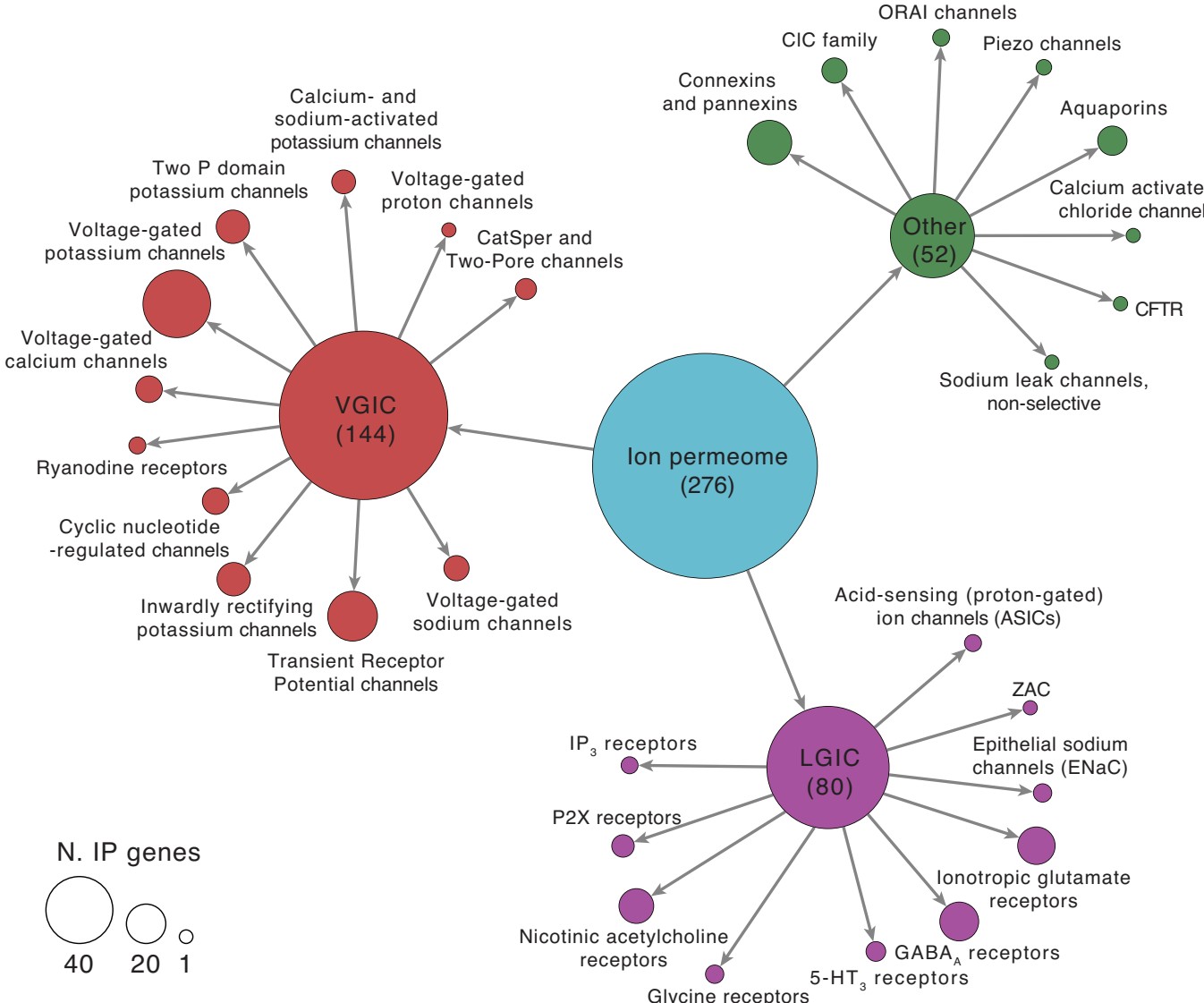

**Figure EV1. Classification of ion permeome (IP) genes used in our study.**

Genes encoding IP proteins with known specific inhibitors were obtained from the Guide to Pharmacology database and filtered to include genes with expression profiles in TCGA. The network shows a hierarchical classification of IP proteins by type or family (nodes) with arrows indicating subclasses derived from larger classes. Node size reflects the number of genes within each type or family. Node color reflects major IP gene classes. Other ion channels (IC), voltage-gated ICs (VGIC), and ligand-gated ICs (LGICs) represent the major classes shown. Source data are available online for this figure.

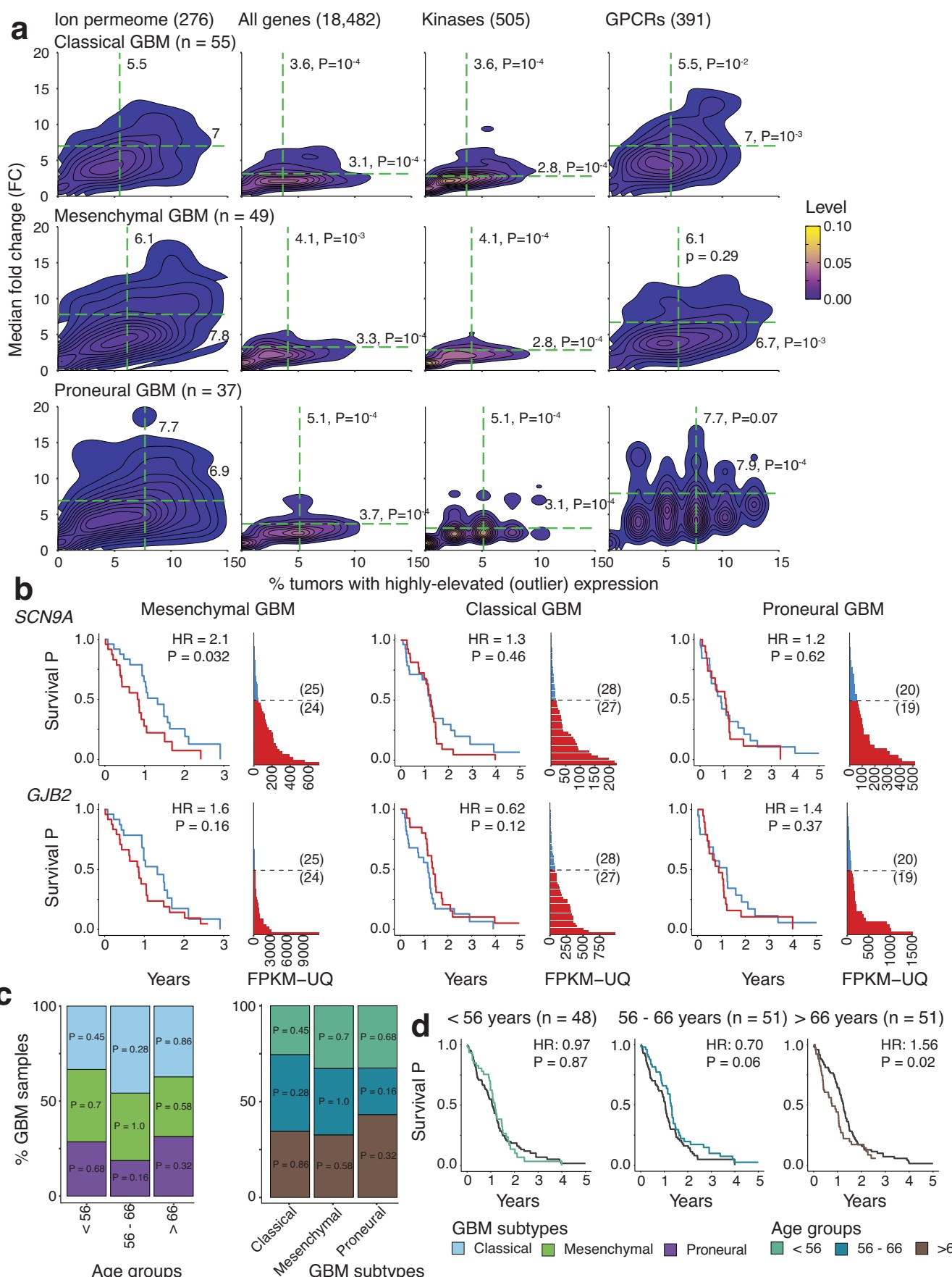

◄ **Figure EV2. GBM subtype analysis of IP gene expression and patient survival.**

(A) High upregulation of IP genes in classical, mesenchymal, and proneural subtypes of GBM. Density plots show the joint distribution of gene expression increase (fold-change (FC), log2) and the fraction of affected cancer samples. IP genes (left) are compared to three controls (middle to right): (i) all protein-coding genes, and two classes of drug targets: (ii) kinases and (iii) GPCRs, with down-sampling performed similarly to the main analysis. Dashed green lines show median values. Representative iterations with median fold-change are shown. (B) Kaplan-Meier plots of overall survival (OS) in mesenchymal (left), classical (middle), and proneural (right) GBMs grouped by *GJB2* or *SCN9A* expression in TCGA. Bar plots show gene expression levels in risk groups. Risk groups were determined by median dichotomisation for *GJB2* and *SCN9A* expression levels. Wald *P*-values, Univariate HR values, and sample counts are shown. Patient age was included as a covariate in TCGA analyses. (C) GBM subtype associations with patient age. Stacked bar plots show the fraction of samples from GBM subtypes (left) or associated ages of patients (right) in each age group or GBM subtype. Mann-Whitney U-test *P*-values are shown. (D) Kaplan-Meier plots of OS in all TCGA GBM patients grouped by patient age. Patient groups were determined as three equally-sized groups based on low, medium, or high age. Wald *P*-values, Univariate HR values, and sample counts are shown. Source data are available online for this figure

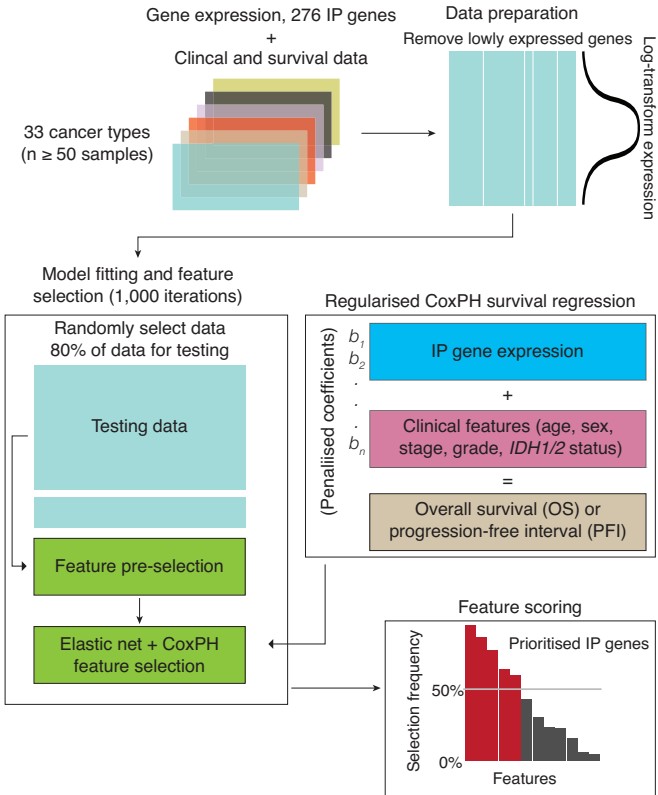

**Figure EV3.  Overview of the machine learning pipeline for identifying survival-associated IP genes.**

IP gene expression was evaluated for significant survival associations individually for 33 cancer types from TCGA. IP gene expression values were log-normalized and used as features in a machine learning framework. Regularized CoxPH models were trained on iterations of 80% of samples within each cancer type, using IP gene expression values and clinical variables (patient age and sex, tumor stage, grade, and *IDH1/2* mutation status) as model features and patient survival as the response variables. Within each iteration, features were first pre-selected using univariate CoxPH models trained, and only genes significantly associated with patient survival ($P < 0.1$, Wald test) were selected for the multivariate model. After 1,000 iterations of feature prioritisation, IP genes associated with patient survival in at least 50% of models were selected as high-confidence hits. Source data are available online for this figure

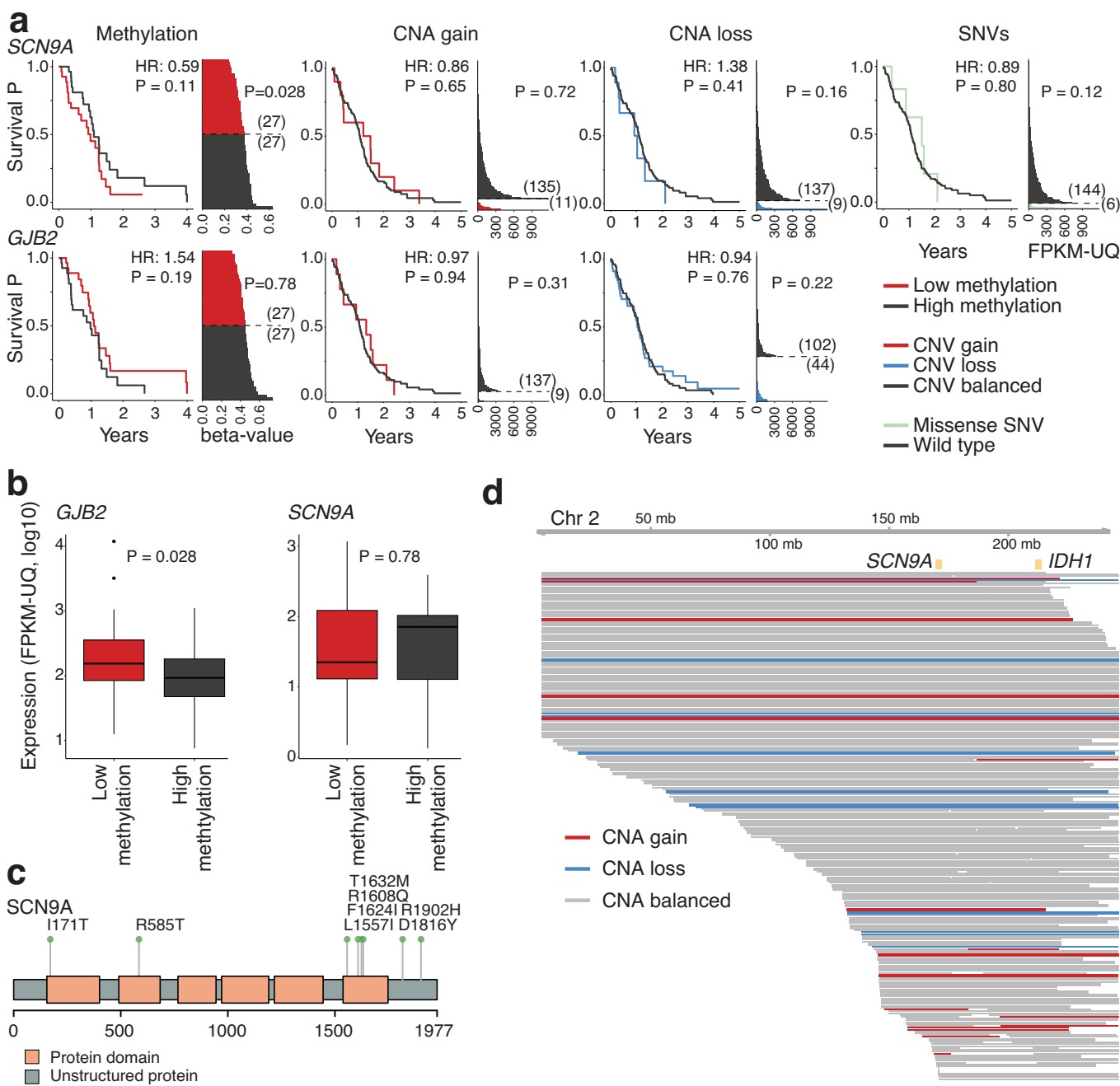

**Figure EV4. Genetic alterations and DNA methylation of *GJB2* and *SCN9A* have few associations with gene expression.**

(A) Kaplan–Meier plots of OS in GBM patients (left to right): *GJB2* and *SCN9A* promoter DNA methylation; relative somatic copy number alterations (CNAs), and somatic single nucleotide variants (SNVs) in GBMs in TCGA. Bar plots show mean gene promoter methylation (left, beta-values) or gene expression in risk groups. Risk groups were determined by median dichotomisation for promoter methylation and presence or absence of a CNAs or SNVs. Wald P-values, Univariate HR values, and sample counts are shown. (B) IP gene expression associations with promoter methylation. Boxplots show gene *GJB2* and *SCN9A* gene expression in samples with high and low gene promoter methylation. Methylation groups were defined by median dichotomisation of mean methylation beta values of each promoter. Box plots span the interquartile range (IQR; 25th-75th percentiles) where median values are shown as lines and whiskers reflect values within 1.5x of IQR. (C) Non-silent SNVs in *SCN9A* in TCGA GBMs. Patients from panel (a) with missense SNVs in *SCN9A* are shown with the reference and alternate amino acids. (D) CNAs in *SCN9A* in GBM associate with CNAs in the adjacent *IDH1*. CNAs for GBM TCGA samples are included with labels showing the genomic loci of *SCN9A* and *IDH1*. Source data are available online for this figure

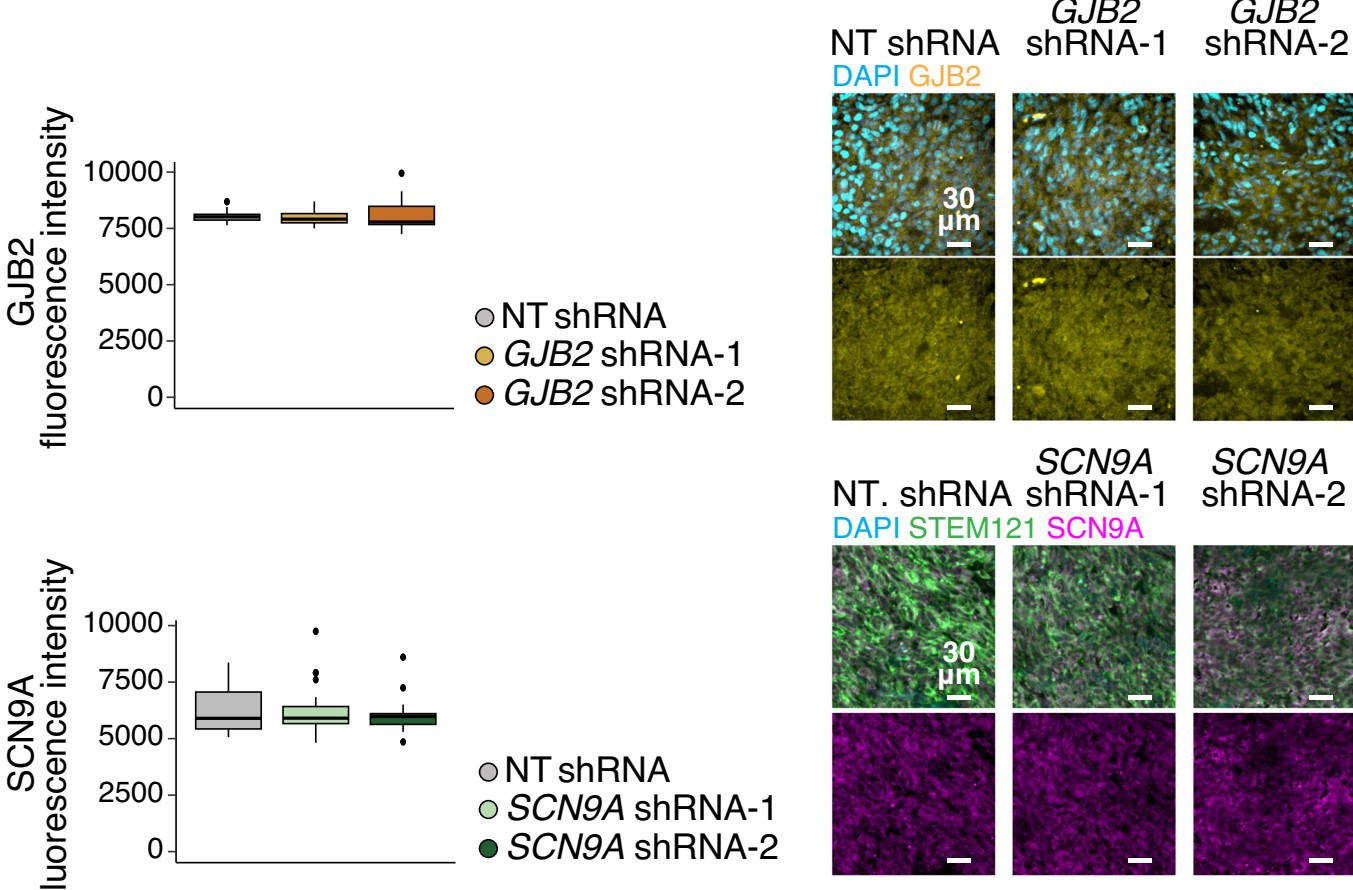

**Figure EV5.  Endpoint GBM tumors in the knockdown groups are shRNA escapers.**

Tumors with knockdown of *GJB2* or *SCN9A* at endpoint have comparable expression of GJB2 or SCN9A. Brains from or IP gene knockdown xenograft mice or NT control xenograft mice were harvested at humane endpoint from tumor outgrowth. Immunofluorescence imaging of GJB2 or SCN9A were performed on *GJB2* or *SCN9A* knockdown tumors, respectively. For GJB2 quantification, DAPI density was used to identify tumor regions because the GJB2 and STEM121 antibodies were both from mouse. For SCN9A quantification, tumor regions were identified by co-staining with human-specific antibody STEM121. Results are from $n = 4$ mice for NT control group, $n = 3$ mice for each *GJB2* knockdown group, and $n = 4$ mice for each *SCN9A* knockdown group. All groups were not significantly different according to Welch's T-tests. Box plots span the interquartile range (IQR; 25th-75th percentiles) where median values are shown as lines and whiskers reflect values within 1.5x of IQR. Source data are available online for this figure

