## [Peer Review File · The EMBO Journal]

Pan-cancer ion transport signature reveals functional regulators of glioblastoma aggression

Alexander Bahcheli, Hyun-Kee Min, Masroor Bayati, Weifan Dong, Alexander Fortuna, Hongyu Zhao, Irakli Dzneladze, Jade Chan, Xin Chen, Kissy Guevara-Hoyer, Peter Dirks, Xi Huang, and Jüri Reimand

DOI: [10.15252/emj.2023114449](https://doi.org/10.15252/emj.2023114449)

Corresponding authors: Jüri Reimand (juri.reimand@utoronto.ca), Xi Huang (Xi.Huang@Sickkids.ca)

Review Timeline:

Submission Date:	5th May 23
Editorial Decision:	23rd Jun 23
Revision Received:	22nd Oct 23
Editorial Decision:	22nd Nov 23
Revision Received:	30th Nov 23
Accepted:	30th Nov 23

Editor: Daniel Klimmeck

Transaction Report:

Dear Dr. Reimand,

Thank you again for the submission of your manuscript (EMBOJ-2023-114449) to The EMBO Journal, as well as for your patience with our response at this time of the year which got protracted due to delayed referee input and discussions in the editorial team. Your study has been sent to two reviewers with expertise in ion transport and brain cancer biology and we have received feedback from both of them, which I enclose below.

As you will see, the referees acknowledge the potential interest and novelty of your comparative analysis, although they also express several issues that will have to be conclusively addressed before they can be supportive of publication of your manuscript in The EMBO Journal. We judge the comments of the referees to be generally reasonable and given their overall interest, we are happy to invite you to revise your manuscript experimentally to address the referees' comments.

As you may have seen on our web page, we generally allow three months as standard revision time. As a matter of policy, competing manuscripts published during this period will not negatively impact on our assessment of the conceptual advance presented by your study. However, we request that you contact the editor as soon as possible upon publication of any related work, to discuss how to proceed. Should you foresee a problem in meeting this three-month deadline, please let us know in advance and we may be able to grant an extension.

When submitting your revised manuscript, please carefully review the instructions below.

Please feel free to approach me any time should you have any questions related to this.

Thank you for the opportunity to consider your work for publication.

I look forward to your revision.

Kind regards,

Daniel Klimmeck

Daniel Klimmeck, PhD
Senior Editor
The EMBO Journal

Instruction for the preparation of your revised manuscript:

- 1) a .docx formatted version of the manuscript text (including legends for main figures, EV figures and tables). Please make sure that the changes are highlighted to be clearly visible.
- 2) individual production quality figure files as .eps, .tif, .jpg (one file per figure).
- 3) a .docx formatted letter INCLUDING the reviewers' reports and your detailed point-by-point response to their comments. As part of the EMBO Press transparent editorial process, the point-by-point response is part of the Review Process File (RPF), which will be published alongside your paper.
- 4) a complete author checklist, which you can download from our author guidelines ([https://wol-prod-cdn.literatumonline.com/pb-assets/embo-site/Author Checklist%20-%20EMBO%20J-1561436015657.xlsx](https://wol-prod-cdn.literatumonline.com/pb-assets/embo-site/Author%20Checklist%20-%20EMBO%20J-1561436015657.xlsx)). Please insert information in the checklist that is also reflected in the manuscript. The completed author checklist will also be part of the RPF.
- 5) Please note that all corresponding authors are required to supply an ORCID ID for their name upon submission of a revised manuscript.
- 6) It is mandatory to include a 'Data Availability' section after the Materials and Methods. Before submitting your revision, primary datasets produced in this study need to be deposited in an appropriate public database, and the accession numbers and database listed under 'Data Availability'. Please remember to provide a reviewer password if the datasets are not yet public (see <https://www.embopress.org/page/journal/14602075/authorguide#datadeposition>). In case you have no data that requires deposition in a public database, please state so in this section. Note that the Data Availability Section is restricted to new primary data that are part of this study.

7) Our journal encourages inclusion of *data citations in the reference list* to directly cite datasets that were re-used and obtained from public databases. Data citations in the article text are distinct from normal bibliographical citations and should directly link to the database records from which the data can be accessed. In the main text, data citations are formatted as follows: "Data ref: Smith et al, 2001" or "Data ref: NCBI Sequence Read Archive PRJNA342805, 2017". In the Reference list, data citations must be labeled with "[DATASET]". A data reference must provide the database name, accession number/identifiers and a resolvable link to the landing page from which the data can be accessed at the end of the reference. Further instructions are available at .

8) At EMBO Press we ask authors to provide source data for the main and EV figures. Our source data coordinator will contact you to discuss which figure panels we would need source data for and will also provide you with helpful tips on how to upload and organize the files.

Numerical data can be provided as individual .xls or .csv files (including a tab describing the data). For 'blots' or microscopy, uncropped images should be submitted (using a zip archive or a single pdf per main figure if multiple images need to be supplied for one panel). Additional information on source data and instruction on how to label the files are available at .

9) We replaced Supplementary Information with Expanded View (EV) Figures and Tables that are collapsible/expandable online (see examples in <https://www.embopress.org/doi/10.15252/emj.201695874>). A maximum of 5 EV Figures can be typeset. EV Figures should be cited as 'Figure EV1, Figure EV2' etc. in the text and their respective legends should be included in the main text after the legends of regular figures.

10) When assembling figures, please refer to our figure preparation guideline in order to ensure proper formatting and readability in print as well as on screen:
<http://bit.ly/EMBOPressFigurePreparationGuideline>

11) For data quantification: please specify the name of the statistical test used to generate error bars and P values, the number (n) of independent experiments (specify technical or biological replicates) underlying each data point and the test used to calculate p-values in each figure legend. The figure legends should contain a basic description of n, P and the test applied. Graphs must include a description of the bars and the error bars (s.d., s.e.m.).

We realize that it is difficult to revise to a specific deadline. In the interest of protecting the conceptual advance provided by the work, we recommend a revision within 3 months (21st Sep 2023). Please discuss the revision progress ahead of this time with the editor if you require more time to complete the revisions.

Referee #1:

This manuscript is a very strong and important contribution to the emerging field of "Cancer Neuroscience". Many findings are intriguing, and match very well to what is known. But more importantly, this manuscript provides many highly interesting roads that can be followed in the future. I am sure this is an important resource for the entire field.

I am not going into more detail, because I have reviewed this manuscript at another occasion before, with a similar assessment that was basically very positive - in the current submission, my few additional recommendations have been optimally addressed by the authors. I have no more recommendations.

Referee #2:

The manuscript entitled Pan-Cancer analysis of the ion permeome reveals functional regulators of glioblastoma aggression by Bahcheli et. provides a computational analysis of gene expression of channels the authors have classified as the ion permeome with in vitro and in vivo validation studies focused on glioblastoma.

The manuscript is interesting and the experimental evaluations are intriguing. However, the manuscript suffers major issues. The largest issue is the authors' fundamental approach of using gene expression with patient survival as the parameters for their machine learning. The field is moving away from the use of overall survival as readouts as so many confounding factors are associated with patient mortality that cannot be uncoupled. For GBM overall survival may still be a valid approach but for the many other cancers analyzed it is not...the one-size fits all approach is not appropriate across all cancers. If the authors want to continue looking at pan-cancer the better metrics would be tumour grade/stage which all that data is available via TCGA or other metrics that truly define disease state and progression on a per-cancer basis. Further to this point, the pan-cancer analysis is not really needed considering the authors spend the majority of the manuscript tailoring this to GBM. I highly recommend the authors focus on GBM and expand on their very interesting in vitro and in vivo findings.

Introduction:

The authors have really glossed over massive amounts of work on ion channels in breast and GI tract cancers and chose to focus on the background of GBM. There have been landmark papers that should be cited and even new papers implicating channels in metastasis, the number one cause of cancer related-death. If the authors decided to continue highlighting the pan-cancer approach then the introduction really needs to be better balanced and representative of pan-cancer not GBM solely.

Results:

1. (Figure 1) I would recommend focusing on GBM specifically. Would it possible to carryout these analyses on different molecular subtypes of GBM? I like this approach, if it could be more focused on GBM and different subtypes it may be more impactful. Are there cancers where there is no clear shift in ion permeome (I think in C but hard to tell with all the data piled up).
2. (Figure 2) This is not very meaningful to the reader. All of this would be better summed up in a table which is easier to read.
3. (Figure 3) The differences in survival based on age group are very intriguing. What does this mean biologically? If you separate based on GBM subtype in the middle-aged group, is there a more pronounced difference in survival of middle-aged overexpressors vs none? Or is there a subtype imbalance in the different age groups?
4. (Figure 3) The scRNAseq needs working. How were the subgroups determined? What genes qualified cells as differentiated vs. stem cell? Further, what's the main difference between the 2 data sets? Why does only one really have all the expected normal cells and the other not? These data need better explanations.
5. I highly recommend assessing CN-alterations, mutations as well as the methylation of SCN9A and GJB2 in GBM.
6. (Figure 4) For in vitro studies, do the shRNA KD cells changes their GBM molecular phenotype or show signs of an altered molecular phenotype transcriptomically? How long after targeting the cells with shRNA do author's carry out phenotypic assessments? Do the authors know the turn-over time of these channels? In part b, there are many nodes presented of which many are not specifically labelled and hard to interpret. Have the authors considered simplifying this to perhaps a larger heatmap where it would be easier to make comparisons between the groups.
7. (figure 6) The author's indicate that the low expression of these makers is more in stem cell-like states while the highest are in differentiated cells. It seems odd to me that as the in vitro data indicates that KD reduces viability so greatly, why would one proceed with a stem cell assay? Did the author's see from their KD profiling that the cells looked more stem cell-like? Did the authors think the two were going to be uncoupled, please clarify that logic in the text. Otherwise, it does not make sense to do the stem cell assay as the cells are dying.
8. (figure 5) Since the KD is reducing viability, couldn't the shortened projections be related to cells dying? Those that don't die, what sort of knockdown do they have? Are the channels expressed more in the filopodia? Are there antibodies that can be used to see subcellular localization of the channels?
9. (Figure 6). The data are really interesting. However, much more indepth analysis is required. This whole figure is based on survival while in the previous figure, the authors discuss nanotubes. There seems to be some missing continuity here. Based on

all the experiments in Figure 5, I would expect the authors to continue down this thought process and at least just look at the tumours to see how they are different and look at the changes they observed in vitro. Simple H&E looking at the cells in the tumours and at the edges. Would the authors see less invasive tumours in the KD (e.g. cells spreading out into the parenchyma?) Do the animals with shRNAs that succumb to the tumours have knockdown or are they all escapers? Also, these models should be used to validate the findings in vitro (e.g. RAC1, RAC2, RAC3, PTCH1).

Discussion:

The parts of the discussion on GBM are good. There is room to expand on this more. The pan-cancer approach is not really highlighted. Again, I strongly suggest the author consider possibly removing this pan-cancer approach and tailoring the manuscript to GBM and expanding on their computational and experimental approaches.

Minor concerns:

1. Imprecise wording: Please do not use words like 'excessive' or 'most samples'. You have the data, please list some quantitative measures.
2. Need to define 'switch-like' to the audience.
3. Clearly define the number of genes analyzed in each class of channels for figure 1.
4. The wording in many of the figures is still small too read and crammed together.
 - a. Figure 4: The networks are too small to read. Also changing color, size and shape is too much in one figure to easily follow. This needs to be reworked and labels changed so they are legible. Please include WB validation of KD if antibodies exist.
 - b. Figure 5: in panel b the WB are sufficient, no need for the expression below.

We would like to thank both reviewers for constructive and positive comments that helped improve our manuscript and strengthen our results. We have addressed all comments. Please find a point-by-point response document below.

Reviewer #1

This manuscript is a very strong and important contribution to the emerging field of "Cancer Neuroscience". Many findings are intriguing, and match very well to what is known. But more importantly, this manuscript provides many highly interesting roads that can be followed in the future. I am sure this is an important resource for the entire field.

I am not going into more detail, because I have reviewed this manuscript at another occasion before, with a similar assessment that was basically very positive - in the current submission, my few additional recommendations have been optimally addressed by the authors. I have no more recommendations.

We would like to thank the reviewer for the very positive comments. We appreciate the opportunity to improve the manuscript based on the reviewer's advice at a previous occasion.

Reviewer #2

The manuscript entitled Pan-Cancer analysis of the ion permeome reveals functional regulators of glioblastoma aggression by Bahcheli et al. provides a computational analysis of gene expression of channels the authors have classified as the ion permeome with in vitro and in vivo validation studies focused on glioblastoma. The manuscript is interesting and the experimental evaluations are intriguing.

We would like to thank the reviewer for the positive note and the constructive comments. Our point-by-point responses are shown below.

However, the manuscript suffers major issues. The largest issue is the authors' fundamental approach of using gene expression with patient survival as the parameters for their machine learning. The field is moving away from the use of overall survival as readouts as so many confounding factors are associated with patient mortality that cannot be uncoupled. For GBM overall survival may still be a valid approach but for the many other cancers analyzed it is not...the one-size fits all approach is not appropriate across all cancers. If the authors want to continue looking at pan-cancer the better metrics would be tumour grade/stage which all that data is available via TCGA or other metrics that truly define disease state and progression on a per-cancer basis.

Thank you for a great comment. Indeed, our models predicted different patient survival readouts based on the recommendations in previous publications of the TCGA consortium (1). Specifically, overall survival (OS) was used for some cancer types and progression-free survival (PFS) was used for others (see **Methods** and **Table EV1**). Also, we incorporated tumor grade and/or stage as covariates in our machine learning models. This is discussed in the **Results** and **Methods** sec-

tions (page 7 and page 24, respectively). For the four candidate genes in GBM, the covariate-adjusted HR values are shown in the dot-and-whiskers plot in **Figure 3**.

Further to this point, the pan-cancer analysis is not really needed considering the authors spend the majority of the manuscript tailoring this to GBM. I highly recommend the authors focus on GBM and expand on their very interesting *in vitro* and *in vivo* findings.

We would like to argue politely that our manuscript is strengthened by the pan-cancer approach as it provides a useful resource of candidate genes to the field (as also emphasized by Reviewer 1). We believe that our systematic analysis of ion permeome genes in more than 30 cancer types is highly relevant to a wide audience of basic and translational researchers who work on these in various experimental projects. Here we focused on GBM, a cancer type of dismal prognosis and strong unmet need, for which we have previous computational and experimental expertise in our group. The candidate genes we found in GBM prompted our interdisciplinary collaboration. Also, we believe that limiting our analysis to GBM and only its subtypes would be also problematic from the point of statistics, since the cohorts are relatively small (see below).

We thank the reviewer for their appreciation and advice on our *in vitro* and *in vivo* work in GBM. To address the comments, we performed several new experiments that are described in detail below.

Introduction: The authors have really glossed over massive amounts of work on ion channels in breast and GI tract cancers and chose to focus on the background of GBM. There have been landmark papers that should be cited and even new papers implicating channels in metastasis, the number one cause of cancer related-death. If the authors decided to continue highlighting the pan-cancer approach then the introduction really needs to be better balanced and representative of pan-cancer not GBM solely.

We thank the reviewer for highlighting the limitations of our literature overview in the introduction and apologise for missing some of the key studies in the field. We have expanded the introduction and now reference several mechanistic studies on the regulation of ion channels in multiple cancer types (breast, GI tract, lung, renal, prostate), and in metastasis (refs (8)-(39) in the manuscript). These studies highlight the functional and clinical impact of the ion permeome in cancer and contextualise the importance of our discovery of novel gene targets. However, given the limited space in our manuscript we politely suggest that further in-depth discussion of the many roles of ion permeome genes in cancer should be left for dedicated review papers.

Results:

1. (Figure 1) I would recommend focusing on GBM specifically. Would it possible to carry out these analyses on different molecular subtypes of GBM? I like this approach, if it could be more focused on GBM and different subtypes it may be more impactful. Are there cancers where there is no clear shift in ion permeome (I think in C but hard to tell with all the data piled up).

We thank the reviewer for this comment. Regarding the first part of the comment, we discuss our rationale to focus the computational prediction of IP genes in all cancer types earlier in this response letter.

Regarding the second part of the comment, we performed new analyses to include three major GBM subtypes where possible (*i.e.*, classical, mesenchymal, and proneural GBMs).

First, we analysed the overexpression of ion permeome (IP) genes in GBM subtypes, comparing all IP genes to protein-coding genes and other major druggable genes (G-protein coupled receptors and kinases) similarly to the main analysis (see new **Figure EV2a** copied below). Generally, we confirmed aberrant expression of IP genes in GBM subtypes significantly exceeds other druggable gene families, consistent with our observations for GBM and the other cancer types we analysed. Of note, the GBM subtype analysis is underpowered statistically due to the small sample sizes we could include (55 classical, 49 mesenchymal, and 37 proneural GBMs were available). These sample sets are close or below our initial sample size cut-off of 50 or more samples that we used in our main **Results (Figure 2)**.

Second, we compared patient overall survival (OS) patterns in GBM subtypes by comparing samples with high and low expression levels of *GBJ2* or *SCN9A*, similarly to our analysis of all GBM samples (see the new **Figure EV2b** copied below). No significant OS differences were found, other than in the mesenchymal group for which high *SCN9A* expression associated with worse prognosis. Only few samples could be analysed and GBM prognosis is dismal regardless of subtypes, so the lack of strong signals is not surprising and can be perhaps attributed to a statistically underpowered analysis with overall poor prognosis apparent in the L-shaped OS curves characteristic of GBM.

2. (Figure 2) This is not very meaningful to the reader. All of this would be better summed up in a table which is easier to read.

We thank the reviewer for this recommendation. We already include a supplementary table that shows this information comprehensively and allows the readers to explore our dataset (Table EV2). That said, we believe that our current visualisation allows the readers to look up specific IP genes while having a broader overview as well. Figure 2a-b show how various candidate IP genes compare to each other and thus we decided to keep the figure format as in our original

manuscript. However, we optimised the figure to include larger gene symbols. We would be happy to continue improving the figure if necessary. In response to the comment, we now provide an additional table to present this data in another format: **Table EV4** describes the data summarised in **Figure 2b**.

3. (Figure 3) The differences in survival based on age group are very intriguing. What does this mean biologically? If you separate based on GBM subtype in the middle-aged group, is there a more pronounced difference in survival of middle-aged overexpressors vs none? Or is there a subtype imbalance in the different age groups?

We appreciate the reviewer’s positive note and constructive comments. We addressed these in our updated manuscript. We separated GBM samples by subtypes and by age groups (**Figure EV3c**) and analysed the survival differences of the different age groups (**Figure EV3d**). No significant associations between GBM subtypes and patient age groups were found, other than the finding that the older age group (patients > 66 years) had significantly worse survival than all other age groups (Hazard Ratio (HR) 1.564, $p = 0.022$). The younger age group (< 56 years) had no significant differences and the middle age group (56 – 66 years) had only sub-significant differences in survival (HR = 0.693, $p = 0.061$). The age groups were roughly equal in size (48-51 samples) and there were no significant differences in GBM subtypes based on age. We note again that these results are based on small sample sizes and therefore not fully reliable. Thus, we added these in our supplementary materials for completeness and did not emphasise these strongly in the main part of the manuscript.

4. (Figure 3) The scRNAseq needs working. How were the subgroups determined? What genes qualified cells as differentiated vs. stem cell? Further, what's the main difference between the 2 data sets? Why does only one really have all the expected normal cells and the other not? These data need better explanations.

We apologise for the lack of clarity. In this analysis we used data from two previous publications where the cell type annotations were reported along with the scRNA-seq data. These included 7,930 cells from IDH1/2 wildtype GBMs (Nefitel *et al.* (2019)) and 55,284 cells from IDH1/2 wildtype and mutant GBMs (Johnson *et al.* (2021)). We then performed UMAP dimensionality reduction on the previously processed gene expression values (TPM) from the two studies. All the cells were labelled using the annotations provided in the original studies. As we used previously published data, the differences in the original experimental designs, data preprocessing steps, and cell type annotation strategies contribute to the differences we observed. We explained this data analysis in the **Methods** section under “*Expression of candidate IPs in GBM at the single-cell level*” on page 26 of our manuscript. We also added a sentence to the **Results** to discuss the differences between the two datasets (page 12).

5. I highly recommend assessing CN-alterations, mutations as well as the methylation of *SCN9A* and *GJB2* in GBM.

We appreciate this comment. When preparing our first submission we systematically analysed the genetic and epigenetic alterations of the two candidate genes in GBM. However, as no striking findings were then apparent, we excluded these data from our manuscript and focused on gene expression patterns instead. As this comment greatly contributes to the completeness of our study, we analysed the data again and added the findings to our manuscript in **Appendix Figure S4**.

First, we investigated the promoter methylation of the two candidate genes in GBM and found no significant associations between promoter methylation of *GJB2* and *SCN9A* and patient survival (**Appendix Figure S4a**, left). Promoter methylation associated with *GJB2* expression but not with *SCN9A* expression (**Appendix Figure S4b**).

Second, we investigated genomic copy number alterations (CNAs) of *GJB2* and *SCN9A* in GBM and found no associations between CNAs and patient survival (**Appendix Figure S4a**, middle). Interestingly, we found that CNAs in *SCN9A* co-occurred with CNAs of the prognostic GBM oncogene *IDH1* due to the genomic proximity of *SCN9A* and *IDH1* such that most CNAs in *SCN9A* also affected *IDH1* and vice versa (19 / 20 of samples or 95%). However, these represent only a minority of the GBM cohort (20/146 or 14%). This is a potentially relevant finding for future studies as *IDH1* is mostly described in the context of SNVs (*i.e.*, the prognostic mutation hotspot IDH1-R132H) rather than CNAs, therefore we added a brief sentence about it in the manuscript.

Third, we studied somatic SNVs and indels in *GJB2* and *SCN9A* in GBM (**Appendix Figure S4a** right, **Appendix Figure S4c**). We found no non-silent mutations in *GJB2* and eight missense SNVs in *SCN9A* in six GBM samples. No significant survival associations were found for the *SCN9A* mutations.

Taken together, the aberrant expression patterns of *GJB2* and *SCN9A* had the strongest associations with GBM patient survival while no consistent signals among the genomic or epigenomic

alterations were found to explain the differential expression or OS characteristics of the two candidate genes.

6. (Figure 4) For in vitro studies, do the shRNA KD cells changes their GBM molecular phenotype or show signs of an altered molecular phenotype transcriptomically? How long after targeting the cells with shRNA do author's carry out phenotypic assessments? Do the authors know the turn-over time of these channels?

Thanks for these great comments. Regarding the first part of the comment, we witnessed transcriptome-wide changes mediated by *GJB2* or *SCN9A* knockdown *in vitro* apparent in the differential expression of 4647 and 2088 genes, respectively (FDR < 0.05; FC > 1.25) (Figure 4a).

RNA-seq was performed on GBM cells at the 4-day timepoint after candidate gene knockdown. Fluorescence imaging was also carried out 4 days after lentiviral shRNA transduction, the viability assay after 7 days, and the limiting dilution assay after 14 days. We included this information in the methods section under “*Cell viability, limiting dilution assay, tunneling nanotube and filopodia imaging*”.

Regarding the turn-over time of these proteins, we observed reduced protein expression and cellular phenotypes 4 days after shRNA treatment (**Figure 5, 6, Appendix Figure S7a**), which indicate that at this time point, GJB2 and SCN9A that were expressed prior to shRNA treatment had been degraded. Therefore, the average turn-over time of these proteins in GBM cells is at most around 4 days.

In part b, there are many nodes presented of which many are not specifically labelled and hard to interpret. Have the authors considered simplifying this to perhaps a larger heatmap where it would be easier to make comparisons between the groups.

We respectfully argue that our “enrichment map” is a preferred visualisation to a heatmap. The major reason is that pathway descriptions are highly redundant and often provide too much information that is difficult to navigate, while the network visualisation provides fewer but more general functional themes that are easier to interpret (these are usually manually curated). In other words, there are too many nodes (*i.e.*, enriched pathways) in the enrichment map of **Figure 4b** to label them all legibly, while our summaries of the sub-networks allow a higher-level overview (most individual labels are removed on purpose as is standard practice for these types of figures; see Reimand *et al.* (2019) (2)).

To address this comment, we have added the full list of detected pathways and their evidence with respect to *GJB2* and *SCN9A* to **Table EV4**. We also added the manual annotations from each pathway in Figure 4b to the Table EV4 so that readers may relate the pathway themes shown in the enrichment map to specific pathways of interest. We reviewed the description of this panel in the figure legend for improved clarity.

7. (figure 6) The authors indicate that the low expression of these makers is more in stem cell-like states while the highest are in differentiated cells. It seems odd to me that as the in vitro data indicates that KD reduces viability so greatly, why would one proceed with a stem cell assay? Did the author's see from their KD profiling that the cells looked more stem cell-like? Did the authors think the two were going to be uncoupled, please clarify that logic in the text. Otherwise, it does not make sense to do the stem cell assay as the cells are dying.

We thank the reviewer for bringing up this important point regarding the limiting dilution assay (LDA) (**Figure 6b-c**). We used LDA to evaluate the proliferative capacity of cells (3,4). Sphere forming frequency was interpreted as the proportion of cells within a population that have the capacity to proliferate and form spheres. As *GJB2* or *SCN9A* knockdown cells have reduced sphere forming frequency, the data indicates that gene knockdowns lead to fewer cells that have the proliferative capacity to form spheres.

To clarify our interpretation of the LDA assay, we performed new immunofluorescence imaging of the mitotic marker phospho-histone H3. We found that *GJB2* or *SCN9A* knockdown led to a

reduction in the total percentage of mitotic cells (**Figure 6d**), demonstrating that *GJB2* and *SCN9A* may promote GBM cell proliferation.

8. (figure 5) Since the KD is reducing viability, couldn't the shortened projections be related to cells dying?

Thank you for highlighting this potential confounding factor. To rule out the possibility that the shortened TNTs were the result of cells undergoing apoptosis, we performed immunofluorescence imaging of the apoptosis marker “cleaved caspase-3” with cell membrane-tagged GFP (mGFP). We observed that *GJB2* knockdown reduced TNT length in cleaved caspase-3 negative, non-apoptotic cells (**Appendix Figure S6a**), suggesting that the shortened TNTs in *GJB2* knockdown cells were not caused by cell morphological changes during apoptosis.

To further show that the shortened filopodia were not associated with apoptotic morphological changes, we repeated the mGFP time-lapse imaging with a longer imaging time. Since the morphological changes associated with apoptosis were previously demonstrated to occur within 2 hours (5,6), we performed time-lapse imaging for 2.5 hours. In cells that did not exhibit apoptotic rounding or blebbing throughout the imaging duration, *GJB2* knockdown resulted in shorter filopodia extension length and lifetime (**Appendix Figure S6b**). These results confirm that the shortened projections in *GJB2* knockdown cells were not due to morphological changes associated with apoptosis.

Those that don't die, what sort of knockdown do they have? Are the channels expressed more in the filopodia? Are there antibodies that can be used to see subcellular localization of the channels?

Thanks for a great question. To address this comment, we obtained antibodies against GJB2 and SCN9A and validated antibody specificity by shRNA KD (**Appendix Figure S7a**). GJB2 and SCN9A were broadly expressed on the GBM cell membrane (**Appendix Figure S7b**). We observed that GJB2 and SCN9A were primarily localised to the soma and not TNTs or filopodia (**Appendix Figure S7b**), suggesting that GJB2 may indirectly affect cellular projections, for example, through a RAC1-associated mechanism.

9. (Figure 6). The data are really interesting. However, much more in-depth analysis is required. This whole figure is based on survival while in the previous figure, the authors discuss nanotubes. There seems to be some missing continuity here. Based on all the experiments in Figure 5, I would expect the authors to continue down this thought process and at least just look at the tumours to see how they are different and look at the changes they observed in vitro. Simple H&E

looking at the cells in the tumours and at the edges. Would the authors see less invasive tumours in the KD (e.g. cells spreading out into the parenchyma?) Do the animals with shRNAs that succumb to the tumours have knockdown or are they all escapers? Also, these models should be used to validate the findings *in vitro* (e.g. RAC1, RAC2, RAC3, PTCH1).

We thank the reviewer for these insightful and helpful questions and the positive note. To complement the *in vitro* data in Figure 5 and establish *in vivo* relevance, we performed xenografts of control and *GJB2* knockdown GBM cells in mice with a time-matched analysis at 12 days post implantation. We observed that RAC1 and pMLC2 (*i.e.*, downstream effector of Rho GTPases) levels were reduced in *GJB2* knockdown tumors (Figure 5e). We analysed tumor invasiveness and found that both sinuosity (*i.e.*, a measure of tumor infiltration) and the sizes of infiltrating tumor colony were reduced in *GJB2* knockdown tumors (Figure 5f). Importantly, filopodia at the invading front were shortened in *GJB2* knockdown tumors as well (Figure 5g). These results validate our *in vitro* findings that *GJB2* is involved in the regulation of RAC1 signaling and cellular projections and demonstrate that *GJB2* regulates GBM invasion *in vivo*.

To address the question of whether the mice in the shRNA groups that did not succumb to the xenografts were due to tumor escapers of the shRNA knockdown, we harvested endpoint tumors and performed immunostaining of *GJB2* or *SCN9A* (Figure EV5). We observed that at endpoint, *GJB2* or *SCN9A* knockdown tumors and control tumors had comparable levels of *GJB2* or *SCN9A* expression, respectively. These results, together with the *in vitro* data showing *GJB2* or *SCN9A* knockdown reduces cell viability and proliferation, suggest that the tumors in shRNA-treated groups at endpoint are escapers.

Discussion:

The parts of the discussion on GBM are good. There is room to expand on this more. The pan-cancer approach is not really highlighted. Again, I strongly suggest the author consider possibly removing this pan-cancer approach and tailoring the manuscript to GBM and expanding on their computational and experimental approaches.

Thanks for the constructive feedback on our discussion section. As indicated previously, we politely argue in favor the pan-cancer approach as a major part of our manuscript. The current state of our manuscript is already quite long. We believe that keeping the discussion succinct is in the best interests of readers and the journal.

We greatly appreciate the following constructive and helpful minor comments that helped improve our manuscript.

Minor concerns:

1. Imprecise wording: Please do not use words like 'excessive' or 'most samples'. You have the data, please list some quantitative measures.

We added specific numbers and quantitative measurements where relevant.

2. Need to define 'switch-like' to the audience.

We provided our definition of 'switch-like' in more detail.

3. Clearly define the number of genes analyzed in each class of channels for figure 1.

We added gene numbers for **Figure 1**.

4. The wording in many of the figures is still small too read and crammed together.

We increased the size of figures to make text more visible and adapted our figures to meet the requirements of *EMBO Journal*.

4a. Figure 4: The networks are too small to read. Also changing color, size and shape is too much in one figure to easily follow. This needs to be reworked and labels changed so they are legible. Please include WB validation of KD if antibodies exist.

We increased the size of the labels within the network of **Figure 4d** to improve legibility. We believe that having 2 shapes in addition to size and color scales is appropriate for this figure.

To validate the knockdown experiments, we obtained antibodies against GJB2 and SCN9A, and performed immunostaining 4 days after lentiviral shRNA transduction in mGFP GBM cells. To quantify GJB2 and SCN9A intensity, we used the mGFP signal to create a region of interest encapsulating the cell membrane. We then measured the fluorescence intensity of GJB2 and SCN9A within this region of interest. We observed that GJB2 or SCN9A intensity was reduced in *GJB2* or *SCN9A* shRNA treated cells, respectively, validating knockdown at the protein level (**Appendix Figure S7a**).

4b. Figure 5: in panel b the WB are sufficient, no need for the expression below.

We appreciate this advice. However, we note that only one of three western blot replicates are shown in the western blot of **Figure 5b**. In addition to these blots, the expression values below the western blots show all three biological replicates of our experiment and display significance. We believe that keeping the expression values for all three replicates is important to demonstrate the statistical significance in our analysis. We reviewed the figure legend for clarity.

Bibliography

1. Cancer Genome Atlas Research, N., Weinstein, J.N., Collisson, E.A., Mills, G.B., Shaw, K.R., Ozenberger, B.A., Ellrott, K., Shmulevich, I., Sander, C. and Stuart, J.M. (2013) The Cancer Genome Atlas Pan-Cancer analysis project. *Nat Genet*, **45**, 1113-1120.
2. Reimand, J., Isserlin, R., Voisin, V., Kucera, M., Tannus-Lopes, C., Rostamianfar, A., Wadi, L., Meyer, M., Wong, J., Xu, C. *et al.* (2019) Pathway enrichment analysis and visualization of omics data using g:Profiler, GSEA, Cytoscape and EnrichmentMap. *Nat Protoc*, **14**, 482-517.
3. Brix, N., Samaga, D., Belka, C., Zitzelsberger, H. and Lauber, K. (2021) Analysis of clonogenic growth in vitro. *Nat Protoc*, **16**, 4963-4991.
4. Ploemacher, R.E., van der Sluijs, J.P., Voerman, J.S. and Brons, N.H. (1989) An in vitro limiting-dilution assay of long-term repopulating hematopoietic stem cells in the mouse. *Blood*, **74**, 2755-2763.
5. Saraste, A. and Pulkki, K. (2000) Morphologic and biochemical hallmarks of apoptosis. *Cardiovasc Res*, **45**, 528-537.
6. Evan, G.I., Wyllie, A.H., Gilbert, C.S., Littlewood, T.D., Land, H., Brooks, M., Waters, C.M., Penn, L.Z. and Hancock, D.C. (1992) Induction of apoptosis in fibroblasts by c-myc protein. *Cell*, **69**, 119-128.

Dear Dr Reimand,

Thank you for submitting your revised manuscript (EMBOJ-2023-114449R) to The EMBO Journal. Please accept my apologies for the unusual delay in assessing your amended manuscript, which was due to protracted referee input as well as detailed discussion in the editorial team. Your revised study was sent back to the referees for their re-evaluation, and we have received comments from both of them, which I enclose below.

As you will see, the experts stated that the work has been substantially improved by the revisions and they are now in broadly favour of publication.

Thus, we are pleased to inform you that your manuscript has been accepted in principle for publication in The EMBO Journal.

We now need you to take care of a number of minor issues related to formatting and data presentation as detailed below, which should be addressed at re-submission.

Please contact me at any time if you have additional questions related to below points.

As you might have noted on our web page, every paper at the EMBO Journal now includes a 'Synopsis', displayed on the html and freely accessible to all readers. The synopsis includes a 'model' figure as well as 2-5 one-short-sentence bullet points that summarize the article. I would appreciate if you could provide this figure and the bullet points.

Thank you for giving us the chance to consider your manuscript for The EMBO Journal. I look forward to your final revision.

Again, please contact me at any time if you need any help or have further questions.

Best regards,

Daniel Klimmeck

>> Please add up to five keywords for your study.

>> Add a 'Disclosure and Competing Interests Statement' to the manuscript.

>> Author Contributions: Please remove the author contributions information from the manuscript text. Note that CRediT has replaced the traditional author contributions section as of now because it offers a systematic machine-readable author contributions format that allows for more effective research assessment. and use the free text boxes beneath each contributing author's name to add specific details on the author's contribution.

More information is available in our guide to authors.

>> Appendix: please provide the appendix as one .pdf file ToC on its first page, including page numbers indicated; Appendix figure legends should be removed from manuscript file, and remain only below the corresponding figures in Appendix PDF.

>> Dataset EV legends: Table EV2 and EV4 should be renamed to Dataset EV1 and EV2 uploaded as individual Datasets file, with legends uploaded as a separate sheet in each Excel file, with the corresponding callouts.

>> Funding information: please update the information provided in our online system. Currently missing: Government of Ontario.

>> Callouts: add a callout for figure 2b in the main text.

>> Source data: the source data checklist needs to be completed.

>> Consider additional changes and comments from our production team as indicated below:

- Figure legends:

1. Please note that the figure legend style does not comply with the journal guidelines i.e. all the figure legends are in a run-on style.
2. Please note that a separate 'Data Information' section is required in the legend of figure 5.
3. "Please define the annotated p values **/* in the legends of figures 6a-b as appropriate.
4. Please indicate the statistical test used for data analysis in the legends of figures 6a-b, d.
5. Please note that information related to n is missing in the legend of figures 2b; 3a, e; 6d; EV4b.
6. Please note that the box plots need to be defined in terms of minima, maxima, centre, bounds of box and whiskers, and percentile in the legends of figures 5c-g; 6d; EV4b

- Table EV1, EV3, EV5-EV7 should be renamed to Table EV1-5 and uploaded as individual Expanded View Content (was Supplementary Information) files, with legends uploaded as a separate sheet in each Excel file, with the corresponding callouts
- EV Table Legends should be removed from manuscript file

We realize that it is difficult to revise to a specific deadline. In the interest of protecting the conceptual advance provided by the work, we recommend a revision within 3 months (18th Feb 2024). Please discuss the revision progress ahead of this time with the editor if you require more time to complete the revisions.

Referee #1:

I recommend acceptance of this relevant and sound work.

Referee #2:

I commend the authors on the very respectful and thoughtful rebuttal. I have thoroughly gone through each of the points. The additional text to the manuscript greatly improved the clarity of the experimental logic and improved the story overall. The additional experiments and data analysis performed strengthen their hypotheses. Further, the high-quality level of experimental investigation done on the channel genes in their GBM models really highlights how their machine-learning approach can identify meaningful biological changes. The paper now has shifted the questioning from what experiments should be done now to support their hypotheses to what experiments could be done in the future. Their work is timely as there is a renewal in the interest in targeting ion channels in cancer. Further, their approach has a broad impact on the field and will be beneficial to many. Well done.

The authors addressed the minor editorial issues.

Dear Dr Reimand,

Thank you for submitting the revised version of your manuscript. I have now evaluated your amended manuscript and concluded that the remaining minor concerns have been sufficiently addressed.

Thus, I am pleased to inform you that your manuscript has been accepted for publication in the EMBO Journal.

Please note that it is The EMBO Journal policy for the transcript of the editorial process (containing referee reports and your response letter) to be published as an online supplement to each paper.

If you do NOT want the transparent process file published, you will need to inform the Editorial Office via email immediately. More information is available here: https://www.embopress.org/transparent-process#Review_Process

On a different note, I would like to alert you that EMBO Press offers a format for a video-synopsis of work published with us, which essentially is a short, author-generated film explaining the core findings in hand drawings, and, as we believe, can be very useful to increase visibility of the work. This has proven to offer a nice opportunity for exposure i.p. for the first author(s) of the study. Please see the following link for representative examples and their integration into the article web page:

<https://www.embopress.org/doi/full/10.15252/emj.2019103932>

Finally, we have noted that the submitted version of your article is also posted on the preprint platform bioRxiv. We would appreciate if you could alert bioRxiv on the acceptance of this manuscript at The EMBO Journal in order to allow for an update of the entry status. Thank you in advance!

If you have any questions, please do not hesitate to call or email the Editorial Office.

Best regards,

Daniel Klimmeck

Daniel Klimmeck, PhD
Senior Editor
The EMBO Journal
EMBO
Postfach 1022-40
Meyerohofstrasse 1
D-69117 Heidelberg
contact@embojournal.org

Submit at: <http://emboj.msubmit.net>